# TGFβ-Smad3 signaling restores cell-autonomous Srsf1-mediated splicing of fibronectin in aged skeletal muscle stem cells

Yuguo Liu[1], Svenja C. Schüler[1], Simon Dumontier[1], Frederic Balg [2], Sonia Bedard [2], Thibaut Desgeorges[3], Jerome N. Feige[3,4], Pierre-Luc Boudreault [1] & C. Florian Bentzinger [1] ✉

Loss of Fibronectin (FN) from the skeletal muscle stem cell (MuSC) niche represents a root cause of regenerative failure in aging. While FN has pleiotropic functions during healthy skeletal muscle regeneration, it remains unclear how aging affects its spatiotemporal specificity for MuSCs. Here, we demonstrate that activated MuSCs secrete an autoregulatory FN splice variant containing the EDB extra domain (EDB(+) FN), which is not expressed by accessory cells in the niche. EDB(+) FN splicing in MuSCs depends on serine/arginine-rich splicing factor 1 (Srsf1) whose promoter is controlled by Smad3. EDB(+) FN knockdown or downregulation in aging affects MuSC proliferation through aberrant integrin signaling and impairs skeletal muscle regeneration. During a defined regeneration interval in aged mice, Smad3 activation using transforming growth factor-beta 1 (TGFβ1) improves MuSC function and skeletal muscle repair by stimulating EDB(+) FN secretion. Altogether, we identify and characterize the TGFβ1-Smad3-Srsf1-EDB(+) FN pathway as a therapeutic target for age-associated regenerative failure.

Sarcopenia, the progressive loss of muscle mass, strength, and function as a consequence of aging, represents a main cause of frailty in older persons[1]. In 2016, sarcopenia received an international classification of diseases 10th revision (ICD-10) code, and it is recognized as a condition linked to increased mortality for which novel therapeutic strategies are urgently needed[2]. In recent years, it became apparent that sarcopenia is accompanied by a major regenerative component leading to impaired healing of skeletal muscle injuries following surgical procedures or accidents[3]. Inactivity due to prolonged bed rest in sarcopenic patients with inefficiently regenerating skeletal muscle wounds further exacerbates protein catabolism and speeds up disease progression.

Muscle stem cells (MuSCs), the principal mediators of skeletal muscle regeneration, become less numerous with age and show an impaired regenerative capacity[3]. Increasing evidence points towards a key role of the MuSC microenvironment in triggering age-associated regenerative failure[4]. Next to cell-cell contacts, paracrine growth factors, and cytokines, MuSCs are regulated by structural extracellular matrix (ECM) proteins providing environmental clues that guide the cells through the different stages of skeletal muscle repair[4–6]. During healthy skeletal muscle regeneration, a transient pro-regenerative fibronectin-rich ECM has been demonstrated to be critical for MuSC function and tissue regeneration[5,7,8]. The transient nature of this regenerative ECM distinguishes it from fibrosis, the chronic deposition of molecules such as collagen I in the interstitial space of skeletal muscle, which can have detrimental consequences, including thickening and stiffening of the tissue[9]. Aging has also been shown to be accompanied by a deregulation of fibro-adipogenic progenitors

[1]Département de Pharmacologie-Physiologie, Institut de Pharmacologie de Sherbrooke, Centre de Recherche du Centre Hospitalier Universitaire de Sherbrooke (CHUS), Faculté de Médecine et des Sciences de la Santé, Université de Sherbrooke, Sherbrooke, QC, Canada. [2]Department of Surgery, Division of Orthopedics, Centre de Recherche du Centre Hospitalier Universitaire de Sherbrooke (CHUS), Faculté de Médecine et des Sciences de la Santé, Université de Sherbrooke, Sherbrooke, QC, Canada. [3]Nestlé Institute of Health Sciences, Nestlé Research, Lausanne, Switzerland. [4]School of Life Sciences, École Polytechnique Fédérale de Lausanne (EPFL), Lausanne, Switzerland. ✉e-mail: cf.bentzinger@usherbrooke.ca

(FAPs), the main ECM producers in skeletal muscle[5,10]. Moreover, young MuSCs seeded onto decellularized ECM isolated from aged muscle increase the expression of fibrogenic markers and show a reduced myogenic capacity[9]. Thus, ECM represents a promising therapeutic target for restoring the myogenic capacity of aged MuSCs.

In spite of a general increase in pro-inflammatory processes and interstitial fibrosis, the biogenesis of certain ECM components is significantly reduced in aged skeletal muscles[10–12]. Our previous work has shown that low levels of fibronectin (FN) in regenerating skeletal muscle in aged mice lead to aberrant adhesion signaling in MuSCs[11]. In aged skeletal muscle, MuSCs are also characterized by reduced levels or mislocalized focal adhesion proteins and lower amounts of activated integrin β1, which is a subunit of one of the main FN receptors[5,13]. Concomitantly, forced activation of integrin β1 using conformation-changing antibodies or injection of full-length FN improves muscle regeneration in aged mice[11,13]. Notably, although it is globally reduced in aged regenerating skeletal muscle, the tissue still contains substantial levels of FN. This observation raises the intriguing question of whether the MuSC microenvironment contains FN isoforms with unique molecular properties that are disproportionately affected by the aging process.

FN is a ~500–600 kDa ECM glycoprotein that forms a dimer linked by C-terminal disulfide bonds[14]. Each FN subunit has a molecular weight of ~230–275 kDa and contains several repeats of domain types I, II, and III, which are composed of anti-parallel β-sheets[5,14]. Human FN is transcribed from a single 75 kb gene containing 47 exons and its pre-mRNA is spliced extensively to create up to 20 isoforms[15–17]. Some of the best characterized splice sites of FN are the extra type III modules (EDA and EDB) as well as the variable V-region between the 14th and 15th type III domain[14,18]. Soluble FN, also known as plasma FN (pFN) is found in the systemic circulation and does not contain the EDA and EDB extra domains. In contrast, cellular FN (cFN), which is secreted by a variety of cells, including fibroblasts, always contains either the EDA, EDB, or both extra domains, and is assembled into an insoluble fibrillar matrix deposited in the vicinity of its source. An insert in the V-region is typically present in cFN but is only found in one of the subunits in the pFN dimer. Variation in *FN* splicing allows for the production of isoforms with different binding properties and biomechanical characteristics that can modify ECM composition during development and regeneration of tissues[14,18,19]. However, despite the diversity of roles associated with FN splice variants in the regeneration of tissues such as the skin and the heart, it remains unknown whether specific isoforms play a role during skeletal muscle regeneration[20,21].

Here we describe the discovery that following activation MuSCs secrete high levels of cFN containing the EDB and EDA extra type III modules into their own niche. Notably, autocrine cFN containing the EDB domain (EDB(+) FN) represents a uniquely specific ECM marker for activated MuSCs, which is not expressed by any other cell type in regenerating skeletal muscle. While other isoforms lack this effect, exposure to EDB(+) FN promotes the proliferation of MuSC-derived myoblasts. Moreover, knockdown of EDB(+) FN but not EDB negative (EDB(−)) FN isoforms phenocopies aging features, including impaired skeletal muscle regeneration and reduced integrin β1 activation in MuSCs. The capacity of MuSCs to produce EDB(+) FN is mediated by serine/arginine-rich splicing factor 1 (Srsf1) whose promoter is activated by Smad3 signaling. Despite being a major contributor to FN synthesis in regenerating skeletal muscle, FAPs contain low levels of phosphorylated Smad3 and do not express EDB(+) FN or *Srsf1*. Aging reduces both EDB(+) FN and *Srsf1* expression in proliferating MuSCs. Moreover, transient induction of Smad3 signaling through timed TGFβ1 application restores Srsf1 levels in aged MuSCs, improves skeletal muscle repair through increased EDB(+) FN secretion and integrin β1 activation, and has no negative short- or long-term effects on interstitial fibrosis. Altogether, our work revealed a previously unrecognized mechanism of autocrine ECM regulation in the MuSC niche

and provides an attractive intervention point for the development of pro-regenerative therapies for sarcopenia.

## Results
### EDB(+) FN is an ECM marker for activated MuSCs
Analysis of single cell sequencing data from homeostatic adult 4-7 months-old C57BL/6 mouse *tibialis anterior* (*TA*) skeletal muscle tissue revealed that muscle stem cells (MuSCs) in quiescence did not express *fibronectin* (FN) transcripts, while it was readily detectable in fibro-adipogenic progenitors (FAPs), the principal ECM-secreting tissue-resident mononuclear cell type (Fig. 1a)[22]. At 5 days post experimental injury (5 dpi), close to the peak of skeletal muscle regeneration, FN was expressed by both MuSCs and FAPs. Conversely, low levels of FN transcripts were observed in immune cells (ICs) at 5 dpi. The mouse mRNA coding for cellular FN (cFN) contains at least one of the two extra type III modules at the EDB and EDA splice sites between exon 24 and 26 (EDB) and exon 32 and 34 (EDA), while plasma FN (pFN) does not contain any of the extra domains (Fig. 1b)[14]. We observed that the EDB and EDA extra domains were largely skipped in FN transcripts from uninjured *TA* skeletal muscle tissue from young 6–8 week-old C57BL/6 mice, but the abundance of both was steeply increased at 5 dpi (Fig. 1c). Relative to total FN mRNA, skeletal muscle tissue contained 160% more EDB and 230% more EDA transcripts at 5 dpi compared to the uninjured condition (Supplementary Fig. 1a, b). Notably, freshly isolated in vitro activated MuSC-derived myoblasts from hindlimb skeletal muscle of young C57BL/6 mice expressed high levels of EDB positive FN (EDB(+) FN), while it was barely detectable in FAPs derived from the same tissue (Fig. 1d). In contrast, EDA positive FN (EDA(+) FN) was expressed at substantial levels by both activated FAPs and myoblasts. Quantification of an amplicon covering the splice-sites of both extra type III modules in the FN mRNA revealed that the majority of transcripts containing EDB also contain the EDA domain in myoblasts (Supplementary Fig. 1c-e). While absolute EDA(+) FN mRNA levels were not significantly changed, EDB(+) FN showed a 43% reduction during myoblast differentiation. However, relative to total FN mRNA levels both EDB and EDA were reduced by 57% and 51% respectively during the differentiation process (Supplementary Fig. 1f, g). Using primers hybridizing to exons 39 and 41 spanning the variable IIICS region of the FN transcript, we determined that FAPs and MuSCs predominantly expressed the 120 amino acid insert (V120) of exon 40 (Supplementary Fig. 1h,i).

We next performed immunostaining for EDB(+) and EDA(+) FN in conjunction with the MuSC membrane marker m-cadherin (M-Cad) in skeletal muscle cross sections of young C57BL/6 mice at 24 hours (h) post injury. This experiment revealed that 91% of total EDB(+) FN and 14% of total EDA(+) FN in the tissue colocalized with M-Cad positive MuSCs (Fig. 1e, f and Supplementary Fig. 1j). Moreover, 2% of EDB(+) FN and 30% of EDA(+) FN staining overlapped with area marked by an antibody detecting all isoforms of FN (panFN) (Fig. 1g, h and Supplementary Fig. 1k). Immunostaining for the endothelial cell (EC) marker CD31 and the IC marker F4/80 in skeletal muscle cross sections of young C57BL/6 mice at 24 h post injury revealed that these cell types did not express EDB(+) FN but showed colocalization with EDA(+) FN (Supplementary Fig. 1l, m). After 42 or 72 h of activation in single skeletal muscle fiber culture, MuSCs produced high levels of both EDB(+) and EDA(+) FN (Fig. 1i,j). Altogether, we observed that activated MuSCs secrete high levels of cFN containing the EDB and EDA extra domains. While EDA(+) FN was also expressed by other cell types in regenerating skeletal muscle, our data revealed that EDB(+) FN represents a remarkably specific ECM marker for activated MuSCs suggesting that it has a distinct regulatory function.

### Autoregulatory EDB(+) FN is required for proliferating MuSCs
We next set out to determine whether the EDB or EDA extra domains have regulatory function for MuSCs. To this end we transfected mouse

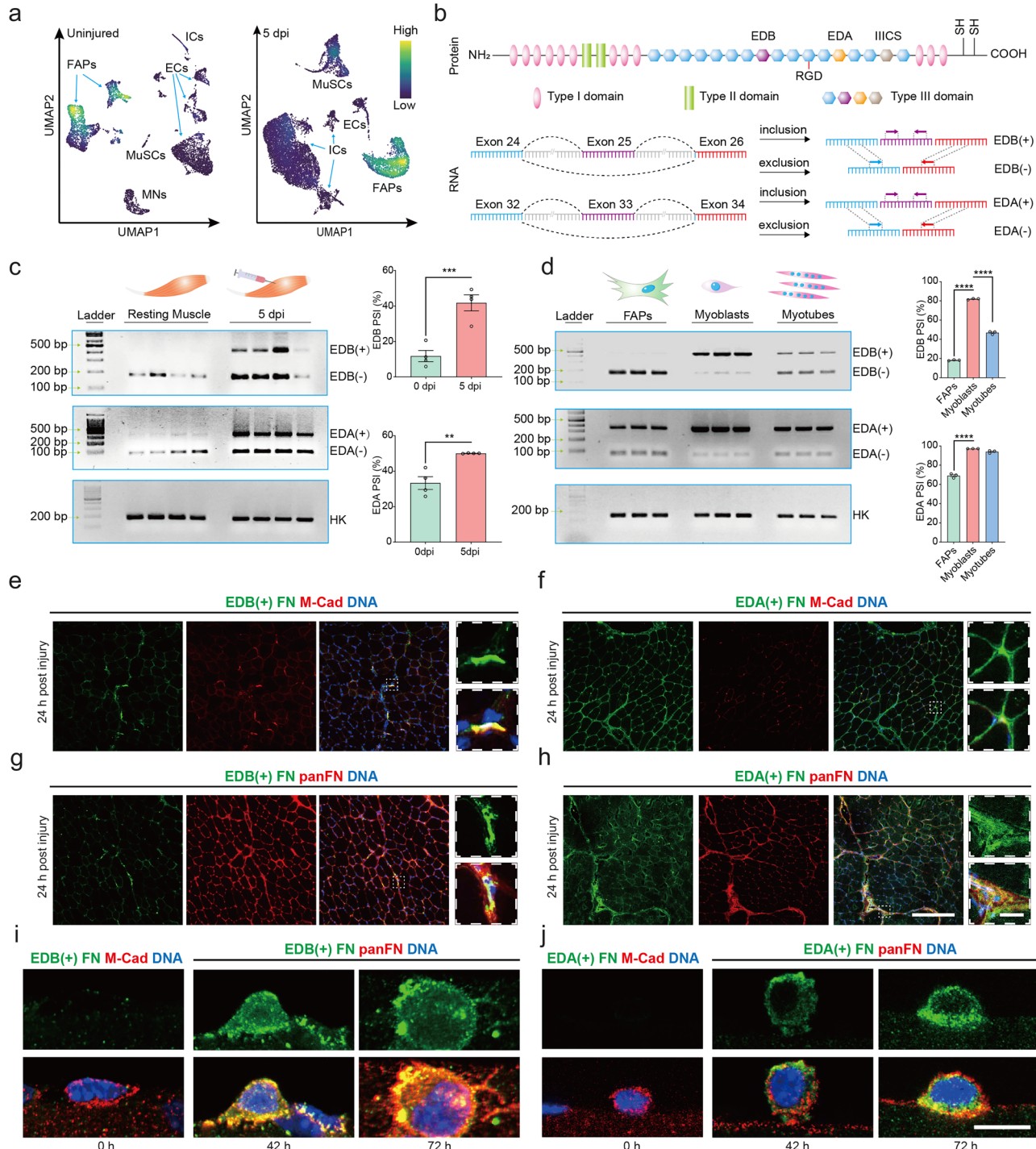

**Fig. 1 | EDB(+) but not EDA(+) FN is enriched in the MuSC niche. a** Uniform manifold approximation and projection (UMAP) analysis of single cell RNA sequencing data from uninjured adult mouse skeletal muscle and at 5 days post injury (5 dpi) overlaid with the gene expression density of total *FN*[22]. FAPs, fibro-adipogenic progenitors; MuSC, muscle stem cells; ICs, immune cells; ECs, endothelial cells; MNs, myonuclei **b** Graphical representation of the mouse FN protein structure and location of the EDA, EDB, and IIICS variable domains, which undergo alternative splicing. Arrows indicate the location of primers used for detection of the respective FN splice variants. **c,d** PCR detection of alternative EDB and EDA splicing of *FN* in uninjured mouse *tibialis anterior* (*TA*) muscles (0 dpi) and at 5 dpi (**c**), and in early passage FAPs, freshly isolated MuSC-derived myoblasts in culture, and differentiated myotubes (**d**) using primers shown in Fig. 1b. The percent spliced in index (PSI) was determined using the formula: in/[in+ex] x 100 (in=exon

inclusion, ex=exon exclusion). Mouse *36b4* (*RplpO*) is shown as a housekeeping (HK) gene. Bars represent means ± SEM from $n = 4$ (**c**) and $n = 3$ (**d**) biological replicates per condition, each from tissues or cells from separate young mice. *P*-values were calculated using a one-tailed Student's *t*-test (c) and one-way ANOVA with Dunnett's post-hoc test (d). ****$p < 0.0001$, ***$p < 0.001$, **$p < 0.01$.
**e–h** Immunostaining of EDB(+) and EDA(+) FN (green) in combination with the MuSC membrane marker m-cadherin (M-Cad, red) or total FN (panFN, red) in *TA* muscle sections of young mice 24 hours (h) post inury. DNA (blue) was counterstained using DAPI. Scale bar = 200 μm (large images) and 20 μm (inserts).
**i, j** Immunostaining of EDB(+) and EDA(+) FN (green) in combination with M-Cad (red) or panFN (red) on single *extensor digitorum longus* (*EDL*) muscle fibers of young mice after 0, 42, and 72 h of culture. DNA (blue) was counterstained using DAPI. Scale bar = 10 μm. Source data are provided as a Source Data file.

fibroblasts with plasmids coding for either EDB(+)EDA(−) FN, EDB(−) EDA(+) FN, or EDB(−)EDA(−) FN (pFN). Immunostaining confirmed that the EDB(+)EDA(−) FN and EDB(−)EDA(+) FN plasmids led to a substantial enrichment of the fibroblast ECM with the respective FN variants (Supplementary Fig. 2a–c). We next co-cultured fibroblasts transfected with the different FN variants with freshly isolated MuSC-derived myoblasts from young mice expressing nuclear zsGreen under the Pax7 promoter[23]. This experiment revealed that ECM containing EDB(+)EDA(−) FN led to a 99%, 52%, and 60% increase in total Pax7-zsGreen positive myoblasts when compared to empty vector (EV), pFN or EDB(−)EDA(+) FN, respectively (Fig. 2a, b). To exclude a confounding effect of cell-cell contacts or soluble factors, we also decellularized the fibroblast cultures after overexpression of the respective plasmids and seeded the ECMs with zsGreen-positive myoblasts (Supplementary Fig. 2d, e). In line with the results obtained from fibroblast co-culture, this experiment revealed a 525%, 275%, and 150% increase in the number of myoblasts in the EDB(+)EDA(−) FN condition when compared to EV, pFN or EDB(−)EDA(+) FN, respectively (Supplementary Fig. 2f). We also observed a 150% increase in myoblast numbers in the EDB(−)EDA(+) FN condition relative to EV.

To interrogate a loss-of-function paradigm, we generated two different siRNAs, each targeting either EDB(+) FN (siEDB(+) FN), or FN that does not contain the respective splice insert (siEDB(−) FN) (Fig. 2c). siEDB(+) FN_a and siEDB(+) FN_b showed a knockdown efficiency of 93% and 92% respectively in myoblasts isolated from hindlimb skeletal muscle of young C57BL/6 mice when compared to a scrambled siRNA (siSCR), while siEDB(−) FN_a and siEDB(−) FN_b led to a reduction of 97% and 59% respectively (Fig. 2d, e and Supplementary Fig. 3a,b). Compared to siSCR, the two siEDB(+) FN led to a 71% and 80% reduction in total myoblast numbers (Fig. 2f, g). The two siEDB(+) FN reduced the percentage of cells positive for the myogenic lineage marker Pax7 by 18% for both siRNAs and the percentage of Ki67 positive proliferating cells by 58% and 38% respectively when compared to siSCR (Fig. 2f–i). In contrast, knockdown with both siEDB(−) FN did not lead to significant changes compared to the siSCR control. We next performed the same experiment using siRNA transfection of MuSCs on single mouse skeletal muscle fibers cultured for 72 h (Fig. 2j). This experiment revealed an 83% and 78% reduction in Pax7 positive MuSCs per clonal cell cluster in the siEDB(+) FN condition compared to siSCR, while again no effect was observed for siEDB(−) FN (Fig. 2k, l).

To determine whether the cell-autonomous effects of EDB(+) FN are conserved, we also tested the knockdown efficiency of two different siRNAs targeting either the human EDB(+) or EDB(−) FN (hFN) mRNA (Supplementary Fig. 3c). siEDB(+) hFN_a and siEDB(+) hFN_b showed a knockdown efficiency of 75% and 70% respectively in early passage myoblasts isolated from healthy young human skeletal muscle when compared to a scrambled siRNA (siSCR), while siEDB(−) hFN_a and siEDB(−) hFN_b led to an mRNA reduction of 48% and 59% respectively (Supplementary Fig. 3d). Resembling its effect in mouse cells, the two siEDB(+) hFN led to a 58% and 55% reduction in total myoblast numbers compared to siSCR, while the two siEDB(−) hFN had no negative effects on the cells (Supplementary Fig. 3e, f). Thus, autoregulatory FN containing the EDB extra domain has conserved and essential functions in promoting the proliferation of activated myogenic progenitors.

### Loss of EDB(+) FN impairs myogenesis in vivo
To study the effects of loss of cFN containing the EDB extra domain during muscle regeneration in vivo, we treated cardiotoxin injured muscles of young C57BL/6 mice with self-delivering variants of the most effective siEDB(+) FN (siEDB(+) FN_a) and siEDB(−) FN (siEDB(−) FN_a) and analyzed them at 5 dpi in comparison to the siSCR control (Fig. 3a). Self-delivering siEDB(+) FN and siEDB(−) FN led to a 59% and 49% reduction in target mRNA levels respectively in muscle tissue

compared to siSCR (Supplementary Fig. 4a, b). Histological analysis using hematoxylin and eosin as well as immunostaining for embryonic myosin heavy chain (eMyHC+) revealed impaired tissue regeneration in the siEDB(+) FN condition, while no effects were observed for siEDB(−) FN and siSCR (Fig. 3b, c). Compared to siEDB(−) FN and siSCR, siEDB(+) FN treatment led to a notable shift of the cross-sectional area (CSA) of newly formed eMyHC+ muscle fibers towards smaller sizes, while their total number per muscle was not affected (Fig. 3d, e). siEDB(+) FN reduced the number of Pax7 positive MuSCs in the tissue by an average of 62% compared to siSCR, while siEDB(−) FN did not lead to significant changes (Fig. 3f, g). These data demonstrate that loss of EDB(+) FN expression leads to regenerative failure accompanied by reduced MuSC numbers and function, phenocopying key features of skeletal muscle aging[3,5,11,13].

### EDB splicing in activated myogenic progenitors requires Srsf1
Bioinformatic screening for consensus sites for alternative splicing factors in the FN coding sequence revealed two GA-rich exonic splicing enhancers (ESEs) motifs for serine and arginine rich splicing factor 1 (Srsf1) localizing to the EDB extra exon (Fig. 4a)[24]. Srsf1 protein contains RNA recognition motifs (RRM) that mediate its interaction with ESEs in pre-mRNAs[25]. Molecular modeling and docking based on the X-ray structure of the Srsf1 RRM domain (PDB ID: 2M8D) showed numerous non-covalent interactions with the 5′-AGGAGAAG-3′ ESE consensus sequence found in the EDB exon (Fig. 4b)[26]. In agreement with a potential role in alternative splicing of FN, the amount of *Srsf1* mRNA was 237% higher in in vitro activated MuSCs than in early passage FAPs (Fig. 4c). To study the effects on alternative splicing of EDB(+) FN, we designed two different siRNAs targeting *Srsf1* (siSrsf1). siSrsf1_a and siSrsf1_b showed a knockdown efficiency of 48% and 49% respectively in myoblasts compared to the siSCR condition (Supplementary Fig. 4c). Notably, siSrsf1_a and siSrsf1_b reduced EDB(+) FN splicing in myoblasts by 72% and 68% respectively, when compared to siSCR (Fig. 4d). In contrast, siSrsf1_a only reduced splicing of EDA(+) FN by 7% (Supplementary Fig. 4d, e). To confirm the role of Srsf1 in an isolated paradigm, we transfected a minigene containing the FN genomic sequence including exons 24 and 26, the EDB extra domain (exon 25), and its flanking introns under control of the *β-actin* promoter into early passage MuSC-derived myoblasts (Fig. 4e)[27]. Using primers specific to the minigene sequence, we observed that increased levels of siSrsf1_a progressively reduced the inclusion of the EDB exon into this portion of the FN mRNA (Fig. 4f and Supplementary Fig. 4f). Conversely, plasmid overexpression of *Srsf1* increased the inclusion of the EDB exon in the FN minigene by 85% compared to empty vector in mouse fibroblasts (Fig. 4g).

We next tested two different siRNAs for their knockdown efficiency of Srsf1 in human myoblasts (hSrsf1). Both si-hSrsf1_a and si-hSrsf1_b reduced hSrsf1 mRNA levels in early passage human myoblasts by 93% compared to the siSCR control (Supplementary Fig. 4g). Akin to mouse cells, si-hSrsf1_a and si-hSrsf1_b reduced inclusion of the EDB extra domain in the hFN mRNA by 83% and 82% respectively (Supplementary Fig. 4h, i). Altogether, these data support the notion that Srsf1 is a potent regulator of EDB extra domain splicing in the FN mRNA in myogenic progenitors in mice and humans (Fig. 4h).

### Smad3 controls Srsf1 and EDB(+) FN expression in activated myogenic progenitors
To identify potential regulatory elements, we analyzed the genomic region -1 kb upstream of the *Srsf1* promoter for transcription factor binding sites. This experiment revealed the presence of a Smad5 binding site and a highly conserved recognition motif for Smad3 (Fig. 5a, b). Chromatin immunoprecipitation using phospho-specific antibodies and three different primer pairs covering the respective region in the *Srsf1* promoter revealed binding of p-Smad3, but not p-Smad5 (Fig. 5c). Antibody labeling detecting p-Smad3 showed 184%

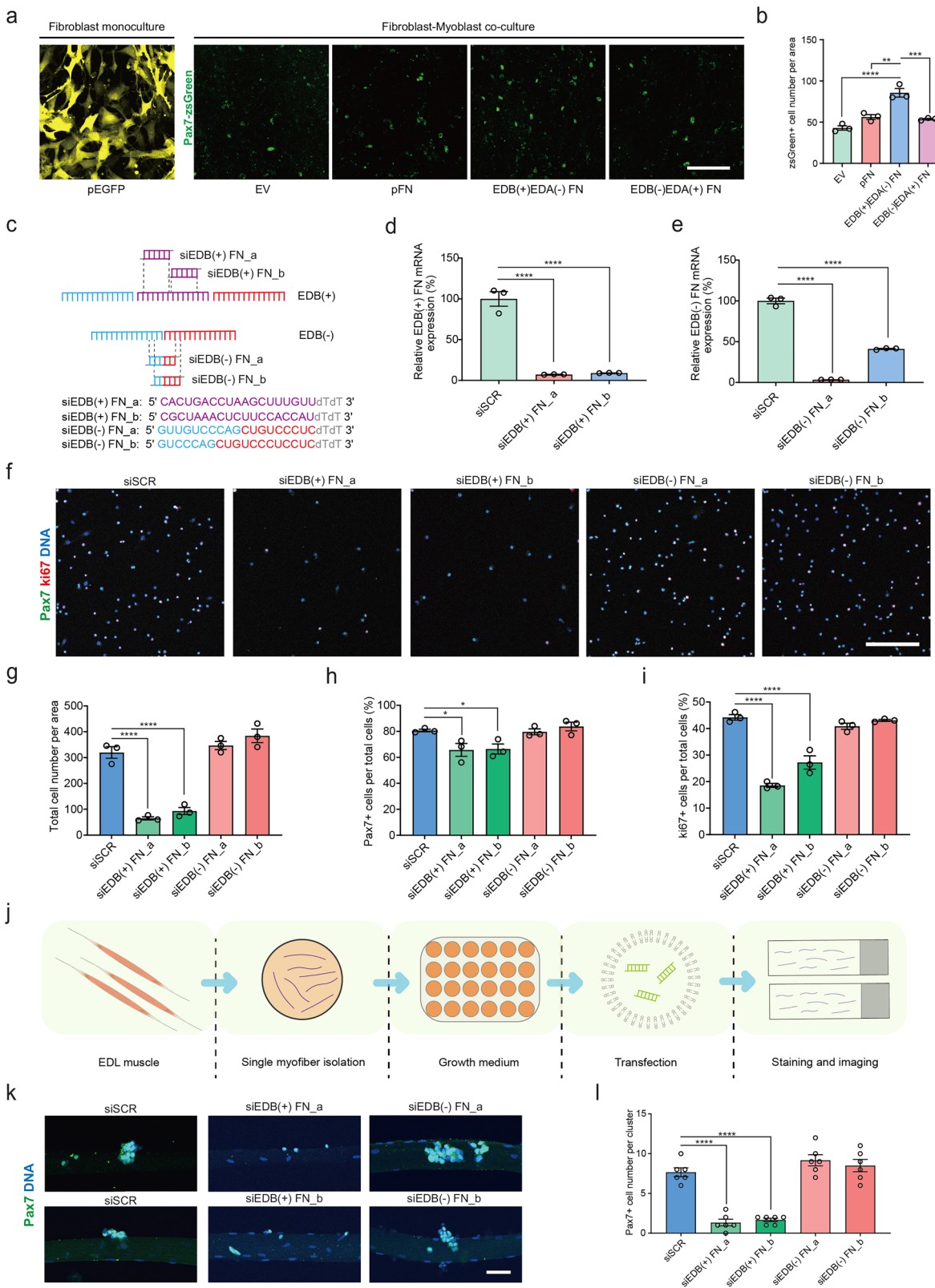

more staining intensity in in vitro activated freshly isolated MuSCs when compared to early passage FAPs (Fig. 5d, e). In agreement with our expression analysis, p-Smad3 staining intensity was associated with 126% higher amounts of Srsf1 protein in MuSCs compared FAPs (Fig. 5d, f). We next used SIS3, a cell-permeable inhibitor of TGFβ1-dependent Smad3 activation and assessed mRNA levels of *Srsf1* in in vitro activated freshly isolated MuSC-derived myoblasts[28] (Fig. 5g).

This experiment revealed that a 48 h incubation of myoblasts with SIS3 reduced *Srsf1* and EDB(+) FN expression in a dose-dependent manner (Fig. 5h, i). However, at 5 nM, the highest assessed SIS3 concentration, we observed that EDB(+) FN splicing in myoblasts was merely reduced by 34%. To assess whether higher doses of Smad3 inhibitor or a longer incubation time would lead to more pronounced effects on EDB(+) FN splicing, we also tested the efficacy of 30 nM

**Fig. 2 | Cell-autonomous EDB(+) FN is essential for the regulation of MuSC function. a**, **b** Representative images of live-imaging of MuSC-derived myoblasts expressing Pax7-zsGreen after 48 h of co-culture with mouse fibroblasts transfected with empty vector (EV), pFN (EDB(−)EDA(−) FN), EDB(+)EDA(−) FN, and EDB(−)EDA(+) FN and quantification of total cell numbers. A GFP plasmid (pEGFP, yellow) is shown to illustrate transfection efficiency. Scale bar = 200 μm. **c** Illustration showing isoform-specific siRNA targeting for knockdown of EDB(+) FN and EDB(−) FN. Two different siRNAs with the suffix _a or _b were designed for each condition. **d**, **e** Semi-quantitative PCR for EDB(+) and EDB(−) FN expression with primers located in the junction region of exon 24 and 26 after isoform-specific siRNA knockdown in MuSC-derived myoblasts compared to the scrambled control (siSCR). **f–i** Representative images and quantification of siEDB(+) FN, siEDB(−) FN,

and siScrambled (siSCR) transfected MuSC-derived myoblasts stained for Pax7 (Pax7+, green) and Ki67 (Ki67+, red) after 48 h of culture. DNA (blue) was counterstained using DAPI. Scale bar = 200 μm. **j** Schematic showing the workflow for siRNA treatment of single *EDL* mouse muscle fibers with their associated MuCSs. **k**, **l** Representative images and quantification of the number of Pax7+ MuSCs (green) per clonal cluster on single *EDL* fibers after siEDB(+) FN, siEDB(−) FN, and siSCR treatment after 72 h of culture. DNA (blue) was counterstained using DAPI. Scale bar = 50 μm. Bars represent means ± SEM from myoblasts or single fibers isolated from $n = 3$ (**b**, **d**, **e**, **g–i**) or $n = 6$ (**l**) biological replicates per condition, each from separate young mice. *P*-values were calculated using one-way ANOVA with Tukey's (**b**) or Dunnett's (**d**, **e**, **g–i**, **l**) post-hoc test. ****$p < 0.0001$, ***$p < 0.001$, **$p < 0.01$, * $< 0.05$. Source data are provided as a Source Data file.

SIS3 after 72 h of incubation. This revealed a 68% reduction of *Srsf1* mRNA levels that was accompanied by a 69% reduction of EDB(+) FN levels (Supplementary Fig. 5a, b). Resembling the effects of EDB(+) *FN* knockdown, increasing concentrations of SIS3 in the medium of freshly isolated in vitro activated MuSC-derived mouse myoblasts progressively reduced the number of total, Pax7 and Ki67 positive cells (Fig. 5j–m).

We next set out to study the effects of SIS3 on *Srsf1* expression and EDB(+) FN splicing in human myoblasts. Compared to the vehicle control, a 48 h incubation with 20 nM SIS3 led to a 72% reduction in hSrsf1 mRNA levels and 90% lower levels of EDB(+) hFN splicing (Supplementary Fig. 5c, d). Resembling its effect on mouse cells, SIS3 reduced human myoblast numbers by 87% when compared to the vehicle control (Supplementary Fig. 5e,f). Collectively, these results suggest a conserved role for Smad3 in controlling *Srsf1* and EDB(+) FN expression in activated myogenic progenitors.

### Autoregulatory EDB(+) FN stimulates integrin signaling and myogenic progenitor proliferation

In order to identify receptors and downstream pathways modulated by autoregulatory EDB(+) FN, we performed proteomic analysis of MuSC-derived myoblasts from young C57BL/6 mice after siEDB(+) FN, siEDB(−) FN or siSCR treatment. Principal component analysis (PCA) revealed that biological replicates from all three conditions showed distinct clustering (Fig. 6a). Compared to siSCR, 419 and 62 unique proteins were significantly regulated by siEDB(+) FN or siEDB(−) FN respectively, and 31 proteins showed overlap between the two conditions (Fig. 6b and Supplementary Fig. 6a, b). KEGG pathway analysis revealed that proteins involved in DNA replication and cell cycle-related pathways were mostly downregulated in the siEDB(+) FN condition compared to the siSCR condition (Fig. 6c, d). Suggesting a compensatory response, proteins involved in ECM-receptor interaction and focal adhesion signaling were induced by knockdown of cell-autonomous EDB(+) FN. Interestingly, amongst upregulated proteins in the ECM-receptor interaction pathway in EDB(+) FN knockdown cells, integrin subunits and related ECM components were overrepresented compared to siSCR (Fig. 6e). The β1 subunit is the most abundantly expressed integrin gene in activated MuSCs[5]. Thus, to test whether cell-autonomous EDB(+) FN activates this ECM receptor subunit, we performed immunostaining of MuSC-derived mouse myoblasts treated with siEDB(+) FN or siSCR using an antibody that exclusively recognizes the active ligand-bound conformation of integrin β1[29]. Quantification of fluorescence intensity revealed that siEDB(+) FN led to a 57% decrease in activated integrin β1 in MuSC-derived mouse myoblasts compared to siSCR (Fig. 6f, g). Moreover, while siEDB(+) in wild-type myoblasts led to a dramatic reduction in cell numbers compared to siSCR or siEDB(−) FN (Fig. 2d–i), it had no effect in integrin β1 knockout (Itgβ1−/−) myoblasts (Fig. 6h–j). These results suggest that cell-autonomous FN containing the EDB extra domain alone, or in combination with EDA, activates integrins in myogenic progenitors and facilitates cell cycle progression through focal adhesion signaling.

### Transient TGFβ1 signaling restores Srsf1 and EDB(+) FN expression in aged MuSCs

To investigate a potential link to aging, we analyzed the amount of *Srsf1* and EDB(+) FN mRNA in skeletal muscle tissue of 22–25-month-old aged mice at 5 dpi and compared them to young control animals between 6–8 weeks of age. This experiment revealed a 44% and 58% reduction of *Srsf1* and EDB(+) FN, respectively in aged tissue compared to the young condition (Fig. 7a, b). In agreement with previous work, we also observed that activated integrin β1 was reduced by 83% in M-Cad positive MuSCs in aged mice compared to young controls (Fig. 7c, d)[13]. It has been shown that in skeletal muscle, TGFβ1 and Smad3 signaling are transiently induced in early stages of regeneration and activate the expression of pro-myogenic ECM genes, while they are downregulated during MuSC differentiation and muscle fiber maturation[5,30]. To test whether it is able to activate Smad3 and induce EDB(+) FN splicing, we treated young mouse myoblasts with TGFβ1 and compared them to vehicle, or epidermal growth factor (EGF) as an unrelated growth factor control. Compared to the vehicle control, TGFβ1 increased Smad3 phosphorylation and EDB(+) FN levels by 29% and 61% respectively, while no increase in either readout was observed for EGF (Supplementary Fig. 7a–c). These results demonstrate that TGFβ1 is able to stimulate the Smad3-Srsf1-EDB(+) FN pathway in MuSC-derived myoblasts derived from young mice.

In aging TGFβ1 levels are chronically increased and have been linked to fibrotic interstitial ECM deposition[31]. Thus, based on the assumption that Smad3 signaling is already partially active in aged MuSCs, we speculated that the cells may be refractory to physiological TGFβ1 levels during tissue regeneration. Thus, to test whether exogenously delivered TGFβ1 is able to reach threshold levels that can restore transient Smad3 signaling to mobilize pro-myogenic ECM synthesis and EDB(+) FN expression, we injected aged muscle at 1 dpi with a single dose of TGFβ1 (Fig. 7e). This protocol led to a 66% increase in *Srsf1* at 2.5 dpi and a 263% increase in EDB(+) FN expression in regenerating muscle tissue at 5 dpi compared to the vehicle control (Fig. 7f, g). TGFβ1-mediated stimulation of EDB(+) FN expression in aged muscle tissue also led to a 201% increase in integrin β1 activation in M-Cad positive MuSCs when compared to the vehicle control (Fig. 7h, i). Compared to vehicle, TGFβ1 injection improved tissue architecture in aged regenerating skeletal muscles and led to larger fibers (Fig. 7j-l). Moreover, transient TGFβ1 treatment increased the number of Pax7 positive MuSCs by 89% in aged muscles at 5 dpi compared to the vehicle control (Fig. 7m, n). Surprisingly, TGFβ1 treated skeletal muscle contained 33% lower levels of interstitial collagen I compared to the vehicle control (Fig. 7o, p). Moreover, when compared to vehicle, the number of interstitial cells typically associated with fibrotic changes was reduced by 35% in TGFβ1-treated muscles (Fig. 7m, q). To determine whether this protocol leads to negative long-term effects on fibrosis or MuSC numbers, we injected aged muscle at 1 dpi with a single dose of TGFβ1 and analyzed the tissue after 30 days (Supplementary Fig. 8a). We observed that tissue architecture, muscle fiber size, collagen I content, and MuSC numbers were not significantly different from the vehicle control

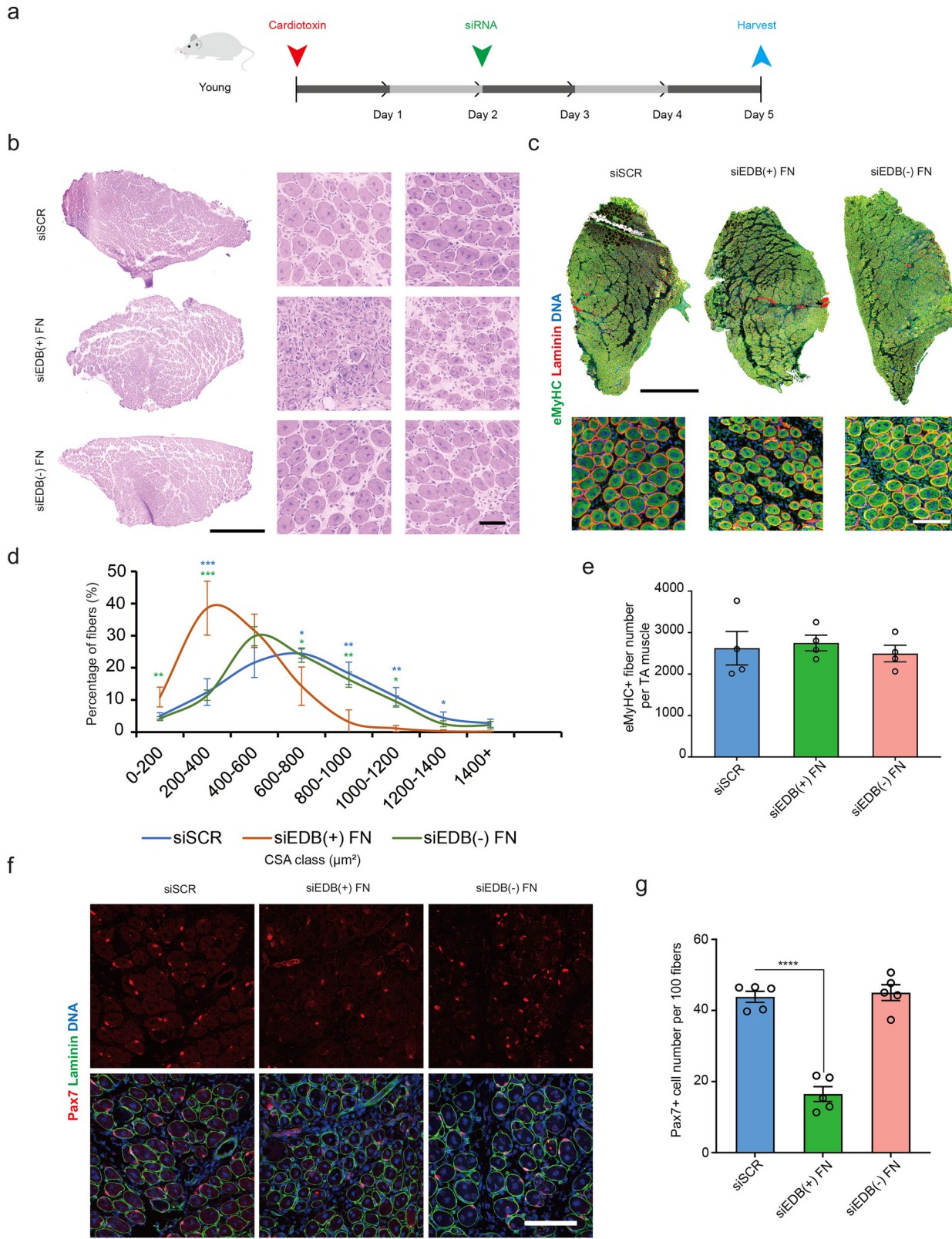

(Supplementary Fig. 8b-g). These results demonstrate that timed transient induction of Smad3 signaling with TGFβ1 is able to mobilize aged MuSCs in the acute phase of muscle regeneration by stimulating *Srsf1* and EDB(+) FN expression without negative short- or long-term effects on the stem cell pool or tissue fibrosis.

Altogether, our study resolved the long-standing question about the relative contribution of different cell types to FN synthesis in the MuSC niche and revealed an intricate autoregulatory mechanism (Fig. 8a). Activated MuSCs secrete cFN containing the EDB and EDA extra domains. EDB(+) FN represents a highly specific ECM marker for the skeletal muscle stem cell niche that is required for integrin activation and cell cycle progression in MuSCs. Inclusion of the EDB extra domain is controlled by the splicing factor Srsf1, whose expression is regulated by TGFβ-Smad3 signaling. Restoration of transient

**Fig. 3 | Loss of EDB(+) FN impairs myogenesis. a** Schematic showing the experimental workflow for in vivo treatement of skeletal muscle with self-delivering siRNAs. **b** Representative hematoxylin and eosin stainings (H&E) of *TA* muscles of young mice treated with self-delivering siSCR, siEDB(+) FN, or siEDB(−) FN. Scale bar = 1 mm for whole muscle sections and 100 μm for enlarged view.
**c** Representative immunostaining for embryonic myosin heavy chain (eMyHC, green), laminin (red), counterstainded for DNA (blue) using DAPI, of *TA* muscles treated with self-delivering siSCR, siEDB(+) FN, or siEDB(−) FN. Scale bar = 1 mm for whole muscle sections and 100 μm for enlarged view. **d** Quantification of the cross-sectional area (CSA) of eMyHC positive (eMyHC+) fibers of *TA* muscles treated with self-delivering siSCR, siEDB(+) FN, or siEDB(−) FN. **e** Quantification of eMyHC+

fibers in TA muscle sections of *TA* muscles treated with self-delivering siSCR, siEDB(+) FN, or siEDB(−) FN. **f, g** Representative immunostaining and quantifcation of for Pax7+ cells (red) in *TA* muscle sections of young mice after treatment with self-delivering siSCR, siEDB(+) FN, or siEDB(−) FN. Sections were counterstained for laminin (green) and DNA (blue) using DAPI. Scale bar = 200 μm. siRNAs termed siEDB(+) FN and siEDB(−) FN (**a-g**) correspond to the siRNAs with the suffix _a described in Fig. 2c. Data points or bars represent means ± SEM from n = 5 (**d, g**) and n = 4 (**e**) biological replicates per condition, each from separate young mice. *P*-values were calculated using one-way ANOVA with Tukey's (**d**) or Dunnett's (**g**) post-hoc test. ****$p < 0.0001$, ***$p < 0.001$, **$p < 0.01$, *$p < 0.05$. Source data are provided as a Source Data file.

---

Smad3 signaling in old skeletal muscles increases *Srsf1* expression and leads to autocrine EDB(+) FN secretion and integrin activation in MuSCs, resulting in improved tissue regeneration without negative effects on fibrosis.

## Discussion

Our study revealed that autocrine secretion of EDB(+) FN is essential for the regenerative function of MuSCs. In skeletal muscle, FAPs residing in the interstitial space, are the main contributors to FN synthesis. However, the laminin-rich basal-lamina physically separates MuSCs and their host muscle fibers from all other tissue compartments, and it appears unlikely that they are in direct contact with interstitial ECM molecules. During skeletal muscle regeneration, the basal-lamina is dynamically remodeled and grows as a function of maturing muscle fibers[5,32]. We observed that differentiating myogenic progenitors downregulate FN expression. However, the basal-lamina surrounding newly formed muscle fibers in early stages of regeneration contains high levels of FN suggesting that MuSCs contribute substantially to remodeling of this ECM structure. In agreement with this notion, MuSCs have also been shown to express a number of laminin isoforms, as well as collagen VI[32,33]. While EDB(+) FN was restricted to the MuSC niche after skeletal muscle injury, we observed that EDA(+) FN was broadly expressed and colocalized with CD31-positive endothelial cells suggesting a role in angiogenesis as described for other tissues[34]. EDA(+) FN has also been implicated in modulating the immune response and activates macrophages, mast cells, and T cells[35–37]. Thus, the roles of FN in skeletal muscle regeneration are pleiotropic and splice variants emerge to be mediators of its spatiotemporal specificity for different cell types during tissue regeneration.

Our data suggests that autoregulatory EDB(+) FN in MuSCs signals through integrin heterodimers containing the β1 subunit and activates downstream pathways involved in cell cycle regulation. Previous work has identified several signaling factors linking integrin activation to cell cycle progression, in particular at the level of the G1/S checkpoint[38]. Integrins have been shown to be upstream of focal adhesion kinase (FAK) and small GTPases, which modulate cyclins and cyclin-dependent kinase inhibitors. Next to integrins, FN binds to a co-receptor complex formed by the heparan sulfate proteoglycan syndecan-4 (Sdc4) and the G protein-coupled receptor frizzled 7 (Fzd7) through which it regulates non-canonical Wnt7a signaling and MuSC self-renewal[7]. MuSCs can divide symmetrically, leading to expansion of the stem cell pool or undergo asymmetric self-renewal, producing progenitors that are committed to differentiation[39]. Recent work has shown that the choice between division modes depends on mitotic spindle orientation relative to the host muscle fiber and the basal-lamina ECM interfacing with the apical and basal poles of MuSCs[40]. Notably, Sdc4, Wnt/Fzd, and integrins have all been linked to mitotic spindle regulation[41–44]. In light of these observations, it is plausible that the spatial distribution of EDB(+) FN in the stem cell niche could regulate MuSC fate decisions by serving as a determinant for spindle pole positioning.

It has been shown that activated integrin β1 is mislocalized and shows reduced levels in aged MuSCs, which goes along with aberrant

changes in several different focal adhesion proteins[13]. Reactivation of integrin β1 using conformation-changing antibodies or injection of full-length FN improves the function of aged MuSCs and stimulates tissue regeneration[11,13]. Our results align well with these observations and revealed that siRNA knockdown or age-induced downregulation of autoregulatory EDB(+) FN leads to an inactivation of integrin β1 signaling in MuSCs. In aged mice, stimulation of EDB(+) FN synthesis using timed and transient TGFβ1 injection increased integrin β1 activation in MuSCs and improved skeletal muscle regeneration. Interestingly, Wnt7a, which together with FN binds to the Fzd7/Sdc4 coreceptor complex, shows over 95% downregulation in aged skeletal muscle tissue, suggesting a link to impaired self-renewal and maintenance of the MuSC pool[7,45]. Altogether, these observations support the notion that multiple age-associated mechanisms affecting MuSC function converge on FN. Direct therapeutic application of ECM molecules such as EDB(+) FN is inherently challenging, as in most cases they are large, poorly soluble, and possess a multimeric nature, which complicates their production, purification, and delivery. By uncovering the conserved TGFβ1–Smad3–Srsf1 splicing pathway and downstream integrin-mediated adhesion signaling as central regulators of cell-autonomous EDB(+) FN in the MuSC niche, we identify actionable targets for therapeutic intervention. These insights lay the groundwork for in vivo modulation of this signaling axis, with the potential to enhance skeletal muscle repair and counteract age-related muscle decline.

Srsf1 belongs to the SR protein family of splicing factors, which contains around a dozen different members in humans[24,46]. The modular domain structure of SR proteins typically contains one or two highly conserved RRMs, as well as a C-terminal RS domain with Arg-Ser dipeptide repeats involved in interactions with other family members. Amongst other functions, Srsf1 interacts with consensus sites in both exonic and intronic sequences in pre-mRNAs and facilitates recognition by the spliceosome. Our experiments revealed that Srsf1-mediated alternative splicing of the FN pre-mRNA leads to inclusion of the EDB extra domain, which is essential for MuSC proliferation. Similar to FN expression, *Srsf1* levels are low in quiescent MuSCs but are induced after activation. In agreement with these observations, constitutive MyoD-specific knockout of *Srsf1* leads to impaired MuSC proliferation and affects muscle mass in mice[47]. We observed that the Srsf1 promoter contains a conserved Smad3 consensus binding site. Smad3 has previously been shown to be required for MuSC proliferation and self-renewal, which would be consistent with a potential role upstream of Srsf1 and EDB(+) FN[48]. Our experiments revealed that Smad3 inhibition led to a dose-dependent reduction Srsf1 and EDB(+) FN expression. Interestingly, we observed that reduced levels of Srsf1 in response to Smad3 inhibition had to reach a certain threshold to affect EDB(+) FN production, which suggests that the involved splicing mechanism is highly efficient in activated MuSCs. Apart from our results regarding EDB splicing in MuSCs, a recent study demonstrated that Srsf1 has a role in regulating inclusion of the FN EDA domain in endometrial fibroblasts[49]. However, we observed that siRNA knockdown of Srsf1 in myoblasts reduced EDA splicing only slightly, while effects on EDB were about ten-fold stronger. Thus, these observations

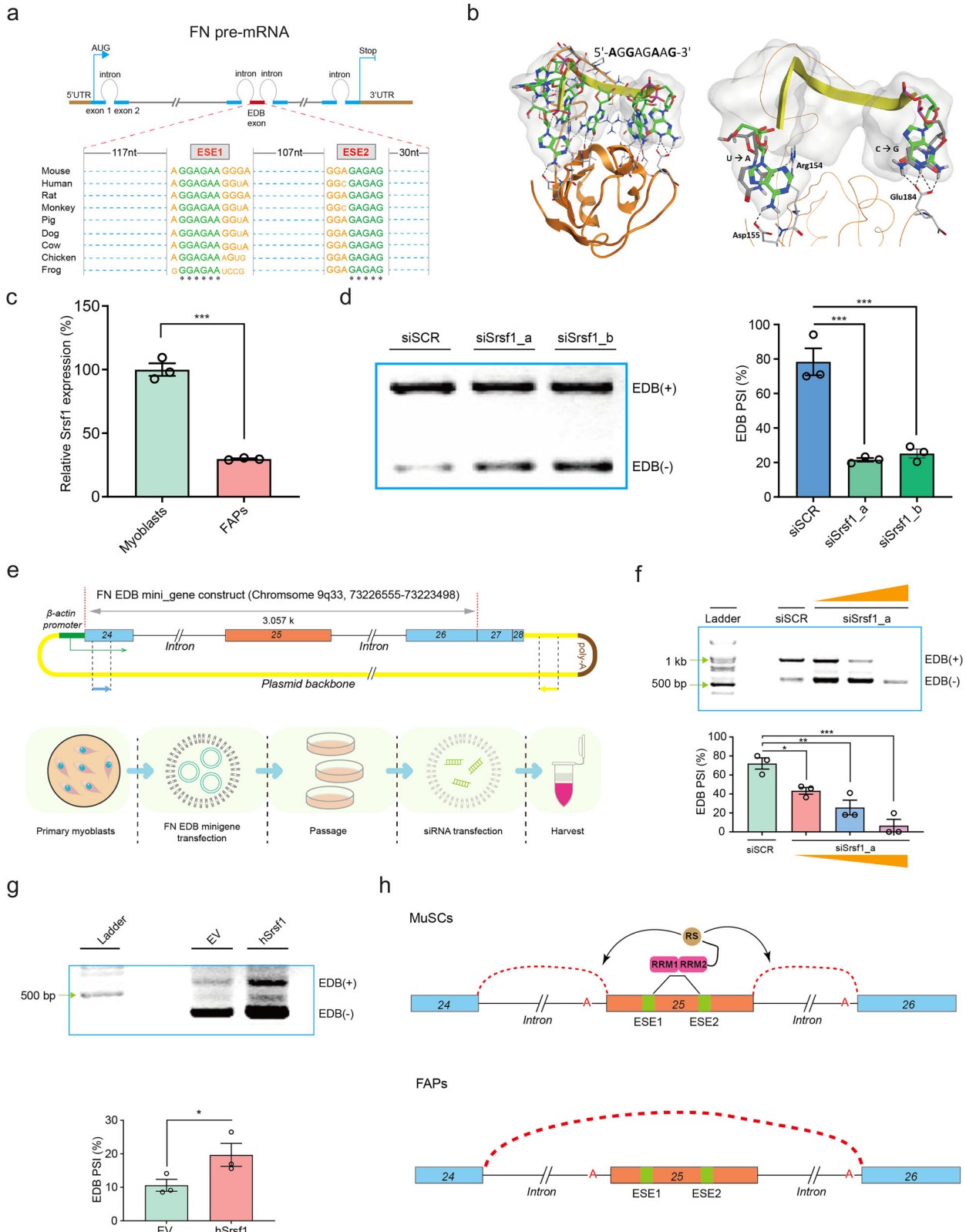

suggest that a factor different from Srsf1 is responsible for the majority of EDA splicing in MuSCs.

Our results show that EDB(+) FN levels in MuSCs are controlled by TGFβ1 signaling, which is a pathway whose chronic activation is recognized to have a central role in triggering fibrosis. In contrast, FN expression in activated MuSCs is part of a generalized augmentation of pro-regenerative ECM synthesis, which is observed across

multiple tissues with the ability to self-repair and appears to be affected as a consequence of the aging process[5,10,50]. Transient pro-regenerative ECM in skeletal muscle differs from chronic fibrosis, which includes an accumulation of fibrillar proteins in the interstitial space, affecting tissue elasticity and impairing stem cell function[5,31,51,52]. The deposition of pro-regenerative ECM in early stages of skeletal muscle regeneration is facilitated by immune cells

**Fig. 4 | Srsf1-mediated alternative splicing promotes EDB exon inclusion.**
**a** Representation of the *FN* pre-mRNA containing two conserved GA-rich exonic splicing enhancers (ESEs) in extra exon 25 coding for the EDB domain and multiple sequence alignment from 9 different vertebrate species. Letter size indicates conservation and asterisks show identical nucleic acids. **b** Left: Molecular docking of the Srsf1 RRM2 (orange) with the ESE1 RNA motif 5′- AGGAGAAG-3′ (green and yellow) showing hydrogen bond interactions. Right: Comparison between the published structure of human RRM2 with 5′-UGAAGGAC-3′ (grey)[26] and the docking results for 5′-AGGAGAAG-3′ (green) showing two major differences in positioning (U → A and C → G). **c** Semi-quantiative PCR analysis of *Srsf1* expression in early passage cultured FAPs and freshly isolated cultured MuSC-derived myoblasts. **d** PCR analysis and PSI quantification of endogenous EDB splicing of the *FN* mRNA in MuSC-derived myoblasts treated with siSrsf1 compared to the siSCR control. Two different siRNAs targeting Srsf1 with the suffix _a and _b were designed. **e** Schematic of the minigene construct containing the extra EDB exon 25 and flanking regions

including exon 24 and 26 and graphical representation of the experimental workflow to assess the effects of *Srsf1* knockdown. Arrows indicate primers used to assess mingene specific transcripts independent from endogenous *FN*. **f** PCR analysis and PSI quantifcation of EDB splicing of the *FN* minigene mRNA in MuSC-derived myoblasts after transfection with increasing amounts of siSrsf1 compared to the siSCR control. **g** PCR analysis and PSI quantification of EDB splicing of the *FN* minigene mRNA in mouse fibroblasts overexpressing *Srsf1* or the empty vector (EV) control. **h** Schematic of the proposed working model for EDB exon splicing. In MuSCs, Srsf1 enhances EDB exon 25 inclusion through RRM binding to GA-rich motifs in the *FN* pre-mRNA. In FAPs, Srsf1 levels are low and the EDB exon 25 is spliced. Bars represent means ± SEM from myoblasts isolated from $n = 3$ biological replicates per condition, each from separate young mice. *P*-values were calculated using a one-tailed Student's *t*-test (c,g) or one-way ANOVA with Dunnett's (d, f) post-hoc test. ***$p < 0.001$, **$p < 0.01$, *$p < 0.05$. Source data are provided as a Source Data file.

releasing a spike in proinflammatory cytokines such as TGFβ1[5,30]. This observation suggests that the immune system could be linked to transient induction of TGFβ1-Smad3-Srsf1 signaling, supporting EDB(+) FN secretion in activated MuSCs. Importantly, the immune system is impaired by the aging process, and systemic levels of proinflammatory cytokines including TGFβ1 are increased. Thus, it is plausible that the aged niche environment leads to chronic activation of Smad3 signaling in MuSCs and desensitizes the pathway, rendering the cells refractory to transient physiological activation during tissue regeneration. We observed that timed injection of TGFβ1 in the earliest stages of skeletal muscle regeneration can overcome this limitation and promote the function of aged MuSCs by upregulating *Srsf1* and EDB(+) FN, which is accompanied by a reduction in interstitial fibrosis.

Although we discovered a novel autoregulatory mechanism controlling MuSC function through EDB(+) FN secretion and characterized the involved up- and downstream pathways, some limitations and outstanding questions of our study remain. The FN molecule contains several different domains with motifs that can engage different types of integrin heterodimers whose activation could have additive or synergistic effects during myogenesis. We observed that almost all EDB(+) FN also contains the EDA module, which raises the possibility that the two domains have essential complementary functions. Loss of FN containing the EDB domain might affect integrin activation in myogenic cells indirectly due to lower availability of binding motifs in constitutive exons or the EDA domain. Furthermore, we observed that the role of EDB(+) FN and its regulation by Srsf1 and Smad3 are conserved in activated human myogenic progenitors in culture. However, experimental paradigms allowing for extrapolation to human skeletal muscle regeneration and aging are limited. Emerging methods such as eccentric contraction-induced skeletal muscle injury or pluripotent stem cell-derived organoids containing MuSCs alongside key niche cell types could inspire future studies about the role of EDB(+) FN during skeletal muscle repair in humans[53–55]. Complementing these biological considerations, our in vitro experiments focusing on EDB(+) FN splicing and related pathways yielded robust effects, ensuring the experiments were adequately powered, and results were validated using complementary reagents and readouts in both mouse and human cells. Since no sex differences were observed, data from male and female mice were pooled. Nonetheless, more subtle sexdependent effects cannot be excluded and may become apparent in studies with greater sample depth.

In summary, our work identified EDB(+) FN as a previously unrecognized ECM-based marker for activated MuSCs. Autoregulatory EDB(+) FN regulates MuSC proliferation through integrins and depends on Smad3 mediated expression of the splice factor Srsf1. The Smad3-Srsf1 axis regulating EDB(+) FN expression is impaired as a consequence of aging but can be reactivated using timed injection of TGFβ1. In summary, our work demonstrates that targeting

autoregulatory EDB(+) FN in the aged MuSC niche holds promise for the development of pro-regenerative anti-aging interventions.

## Methods

### Mice

Husbandry and all experimental protocols using mice were performed in accordance with the guidelines established by the animal committee of the Université de Sherbrooke, which are based on the guidelines of the Canadian Council on Animal Care (ethical protocol 2022-3553). 6–10 week-old young and 22-25 month-old aged C57BL/6 mice were purchased from JAX mice (strain #:000664) or the Quebec Network of Research on Ageing (RQRV). Mice were housed at 22.1–22.3 °C, in 33–44% humidity, with a 12 h light and dark cycle. Experiments were designed to include male and female mice in approximately equal proportions, and animals were randomized into experimental groups. Except for animals that died a natural death during the course of the experiments, no mice were excluded from the study. Unless stated in the figure legends, all experiments were performed using young mice. Reanalyzed single-cell sequencing data were originally generated using 4-7 months-old adult C57BL/6 mice[22].

### Muscle regeneration, in vivo siRNA delivery, and TGFβ1 injection

Muscle injury was induced by injection of 50 μl of 10 μM cardiotoxin (CTX, Latoxan, L8102) in 0.9% NaCl (Stevens, 0195AF01) through the skin into the *tibialis anterior* (*TA*) muscle of isoflurane-anesthetized mice treated with a single dose of buprenorphine (McGill University, Vetergesic multidose) for pain management. For self-delivering in vivo siRNA treatment (Supplementary Table 1), 100 μg Accell siRNA (Dharmacon) in 50 μl 0.9% NaCl (Stevens, 0195AF01) was injected through the skin into the *TA* muscle 2 days post-CTX injury. For TGFβ1 treatment, 150 ng recombinant protein (Biolegend, 763102) in 50 μl 0.9% NaCl (Stevens, 0195AF01) was injected through the skin into the *TA* muscle 24 h post-CTX injury. Following $CO_2$ euthanasia, *TA* muscles were harvested at 5 or 30 days post-injury. For biochemical analysis, muscles were snap-frozen in liquid nitrogen, and for cryosectioning, the tissue was embedded in gum tragacanth (Sigma-Aldrich, G1128) and snap-frozen in liquid nitrogen chilled isopentane (Sigma-Aldrich, M32631).

### Primary myoblast isolation and culture

Primary mouse myoblasts were isolated by mechanical mincing of hind-limb skeletal muscle tissue and digestion using collagenase B (Roche, 11088831001) and dispase (Sigma-Aldrich, D4693). Subsequently, cells in the homogenate were stained with biotinylated anti-Vcam1 antibody (Biolegend,105704) for isolation of MuSCs or with biotinylated anti-PDGFRα (R&D Systems, BAF1062) for FAPs isolation (Supplementary Table 2). After incubation with streptavidin-conjugated microbeads (Miltenyi Biotec, 130-048-101), cells were purified with a midiMACS Separator (Miltenyi Biotec, 130-042-302).

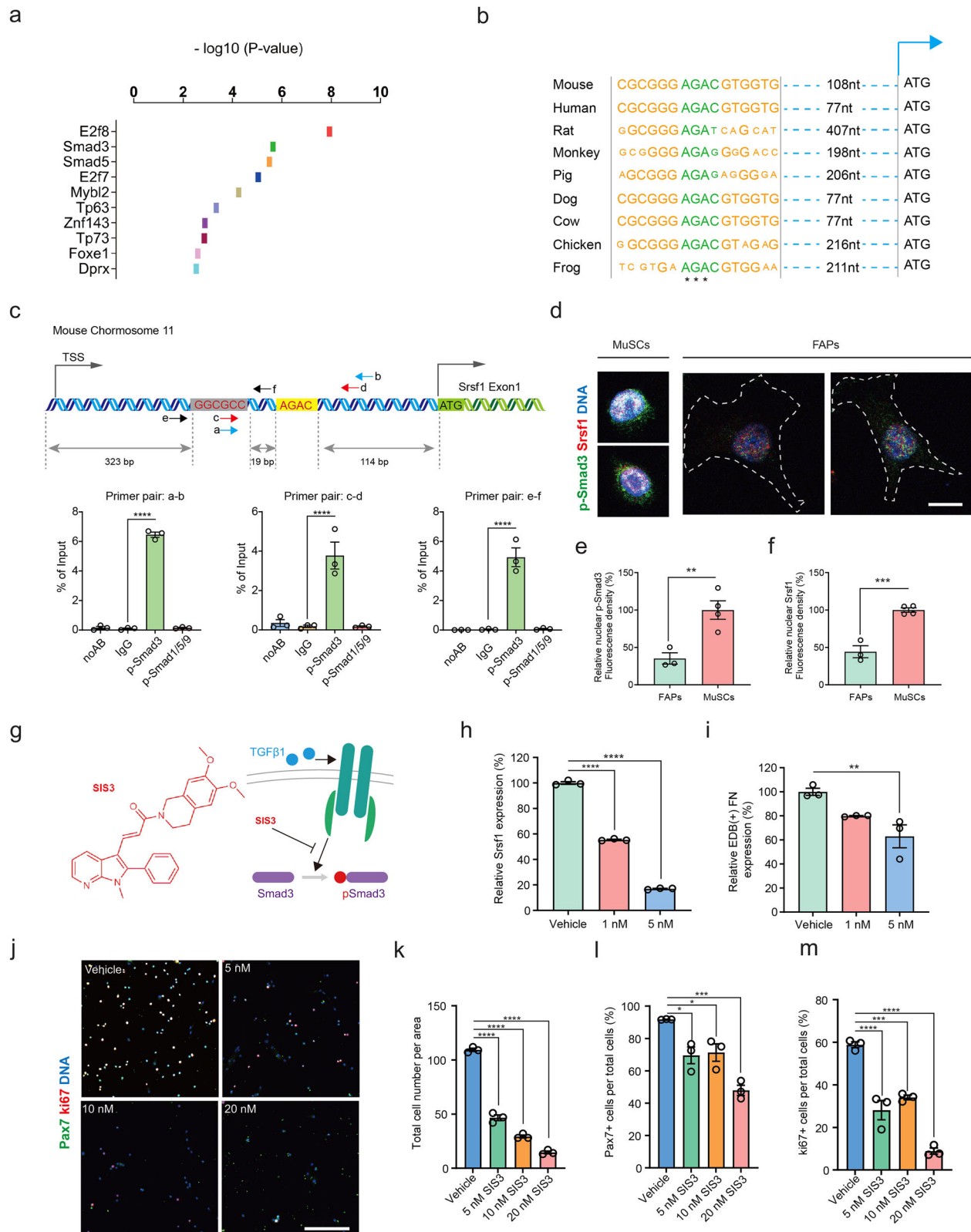

Freshly isolated MuSC-derived myoblasts were cultured in Matrigel (Corning, 47743-178) coated dishes in Ham's F10 (Wisent, 318-050-CL), 20% fetal bovine serum (Wisent, 080450), 1% Penicillin-Streptomycin solution (Wisent, 450-201-EL), and 2.5 ng/mL bFGF (R&D systems, 3139-FB-025). For differentiation, myoblasts at ~80% confluence were switched to DMEM (Wisent, 319-005-CL) with 2% horse serum (Wisent, 065150) for 7 days. FAPs were cultured in DMEM (Wisent, 319-005-CL) with 10% fetal bovine serum (Wisent, 080450) and 2.5 ng/mL bFGF (R&D systems, 3139-FB-025). Primary mouse cells were passaged using 0.25% Trypsin-EDTA (Wisent, 325-043-EL) when reaching a confluence of 60–80%. For all experiments with primary mouse cells, only freshly isolated cells, or cells cultured to a maximum of 2 passages, were used. Media were changed every second day, and all cells were cultured at 37 °C in a 5% $CO_2$ incubator. Primary Pax7-zsGreen mouse myoblasts

**Fig. 5 | Smad3 signaling regulates Srsf1 expression and EDB(+) FN levels.**
**a** Representation of transcription factor binding sites up to 1 kb upstream of the mouse *Srsf1* start codon. The *p*-value shows the probability of obtaining the same interaction score on a random sequence. **b** Multiple sequence alignment of the Smad3 binding site in the *Srsf1* promoter from 9 different vertebrate species. Letter size indicates conservation and asterisks show identical nucleic acids. **c** Schematic of the mouse *Srsf1* promoter around the Smad3 (yellow box, red fonts) and Smad5 (gray box, red fonts) binding motif and chromatin immunoprecipitation (ChIP) using MuSC-derived myoblasts with p-Smad3 and p-Smad1/5/9 antibodies compared to the isoform IgG and no antibody (noAB) controls. ChIP primer pairs in the schematic are indicated by arrows. TSS = Transcription start site. Readouts were normalized to input. **d–f** Representative immunostaining and quantification of nuclear fluorescence intensity of p-Smad3 (green) and Srsf1 (red) in freshly isolated

cultured MuSCs and early passage FAPs from young mice. DNA (blue) was counterstained using DAPI. Scale bar = 10 μm. **g** Schematic showing the molecular structure and mechanism of action of the cell-permeable inhibitor of TGFβ1-dependent Smad3 phosphorylation SIS3. **h,i** Quantification of the effect of increasing concentrations of SIS3 on *Srsf1* and EDB(+) *FN* expression in MuSC-derived myoblasts by semi-quantitative PCR. **j–m** Representative images and quantification of Pax7+ (green), and Ki67+ (red) MuSC-derived myoblasts treated with increasing concentrations of SIS3. DNA (blue) was counterstained using DAPI. Scale bar = 200 μm. Bars represent means ± SEM from myoblasts isolated from *n* = 3 biological replicates per condition, each from separate young mice. *P*-values were calculated using a two-tailed Z-test (**a**), a one-tailed Student's *t*-test (**e, f**), or one-way ANOVA with Dunnett's (**c, h, l, k–m**) post-hoc test. ****$p < 0.0001$, ***$p < 0.001$. **$p < 0.01$. *$p < 0.05$. Source data are provided as a Source Data file.

with nuclear fluorescence were isolated from StemRep dual color reporter mice obtained from Jerome N. Feige[23]. Integrin beta-1 knockout (Itgβ1[-/-]) myoblasts were generated from mice carrying a heterozygous Pax7-CreERT2 (JAX, Strain #:017763)[56] and homozygous floxed Itgβ1 (JAX, Strain #:004605)[57] allele. To induce Cre-mediated recombination and generate Itgβ1[-/-] myoblasts, the cells were treated for 6 days with tamoxifen (Toronto Research Chemicals, T006000-25) at 5 nM final concentration in culture media.

### Primary human myoblast isolation and culture
Human skeletal muscle tissue for primary myoblast isolation was obtained from 3 male and 1 female individual aged between 14 and 26 years. The tissue, obtained with written informed consent from patients, was a by-product of orthopedic knee surgeries and was collected with approval from the Bureau d'Autorisation des Projets de Recherche (BAPR) CIUSSS de l'Estrie – CHUS, Université de Sherbrooke (ethical protocol 2019-2608). Tissues were mechanically minced and digested with collagenase B (Roche, 11088831001) and dispase (Sigma-Aldrich, D4693). Subsequently, myoblasts in the homogenate were stained with CD56 antibody (Thermo Fisher Scientific, BDB555515) (Supplementary Table 2). After incubation with streptavidin-conjugated microbeads (Miltenyi Biotec, 130-048-101), human myoblasts were purified with a midiMACS Separator (Miltenyi Biotec, 130-042-302). Human myoblasts were cultured in Cell+ surface dishes (Sarstedt, 83.390) containing skeletal muscle cell growth medium (Lonza, CC-3245). Primary human myoblasts were passaged using 0.25% Trypsin-EDTA (Wisent, 325-043-EL) when reaching a confluence of 50-70%. For all experiments with primary human myoblasts, only freshly isolated cells, or cells cultured to a maximum of 4 passages, were used. Media were changed every second day, and all cells were cultured at 37 °C in a 5% CO2 incubator.

### Fibroblast-myoblast co-culture
NIH-3T3 mouse fibroblast (Signosis, PC-004) were cultured in polystyrene culture dishes (Sarstedt, 83.3902) containing DMEM (Wisent, 319-005-CL) media supplemented with 10% fetal bovine serum (Wisent, 080450) and 1% Penicillin-Streptomycin (Wisent, 450-201-EL). For expression of FN isoforms the cells were transfected with plasmids (Supplementary Table 3, Addgene, 120401, 120402, and 120403) using lipofectamine 3000 (Thermo Fisher, L3000008) and opti-MEM (Life Technologies, 31985062). An empty pcDNA3.1 plasmid with 3xFlag (Addgene, 182494) was used as an experimental control, and a plasmid expressing EGFP (Addgene, 86776) was used to monitor transfection efficiency. After transfection, the cells were grown to full confluency for approximately 2 days and then switched to myoblast growth medium as outlined above. Primary mouse myoblasts expressing nuclear zsGreen under control of the *Pax7* promoter[23] were then seeded onto the fibroblast mono-layer. After co-culture for 48 h, live cell imaging was performed using a FV1000 laser scanning confocal microscope (Olympus).

### Myoblast culture on decellularized fibroblast ECM
NIH3T3 mouse fibroblast (Signosis, PC-004) were transfected with plasmids encoding different FN isoforms and cultured as described for fibroblast-myoblast coculture. Around 2 days after plasmid transfection, when the fibroblast reached 100% confluency, the cells were washed with DPBS (Wisent, 311-425-CL), and were then treated with 37 °C prewarmed 0.5% Triton-X100 (Sigma-Aldrich, T8787) supplemented with 20 mM Ammonium Hydroxide (Fisher Scientific, A669-500) in DPBS (Wisent, 311-425-CL) for 5 minutes at 37 °C. The resulting fibroblast ECM was then washed with DPBS and incubated with Deoxyribonuclease I (Thermo Scientific, EN0521) for 30 minutes at 37 °C, and washed with DPBS (Wisent, 311-410-CL). Mouse myoblasts expressing nuclear zsGreen under control of the Pax7 promoter[23] were seeded onto the decellularized ECM and cultured for 48 h as described for fibroblast-myoblast coculture.

### Treatment of myoblasts with siRNA, growth factors, and small molecular inhibitors
Cells in culture were generally treated with 4.5 nM siRNA (Supplementary Table 1), except for experiments with multiple doses for which concentrations between 1 and 4.5 nM were used. siRNA transfection was performed using lipofectamine RNAiMAX (Thermo Fisher, 13778030) and Opti-MEM (Life Technologies, 31985062), and the cells were harvested and analyzed at 48 h post-transfection. Growth factors and SIS3 (Supplementary Table 4, Sigma-Aldrich, 566405) were solubilized in a DMSO (Bio Basic, 67-68-5) stock solution. Myoblasts were exposed for 48 h to a concentration of 1-30 nM SIS3 (Sigma-Aldrich, 566405), or 20 ng/ml TGFβ1 (Biolegend, 763102) or EGF (R&D Technologies, 2028-EG) in the culture media.

### Plasmid transfection and FN minigene assay
The FN minigene containing the EDB exon and flanking sequences was made available by Dr. Richard Hynes through the Addgene plasmid repository (7iBi89, Plasmid #14065)[27]. The FN minigene was transfected into primary cells using Lipofectamine 3000 (Thermo Fisher, L3000008) and opti-MEM (Life Technologies, 31985062) according to the manufacturer's instructions and treated with siRNAs after one passage. For detection of the FN minigene transcript and EDB inclusion or exclusion, forward primers were designed to match the plasmid vector sequence, while reverse primers were designed to match exon 26 downstream of the EDB exon (Supplementary Table 5). For the minigene assay, cells were co-transfected with the EDB minigene plasmid (Addgene, 14065) and a plasmid expressing human Srsf1 protein (Addgene, 99021) at 1:2 ratio using Lipofectamine 3000 (Thermo Fisher, L3000008) and opti-MEM (Life Technologies, 31985062). Cells were harvested and analyzed at 48 h post-transfection. For overexpression, NIH-3T3 fibroblasts were transfected with a plasmid coding for Srsf1 and the respective empty vector as outlined above for minigene transfection (Supplementary Table 3, Addgene, #99021 and #182494)

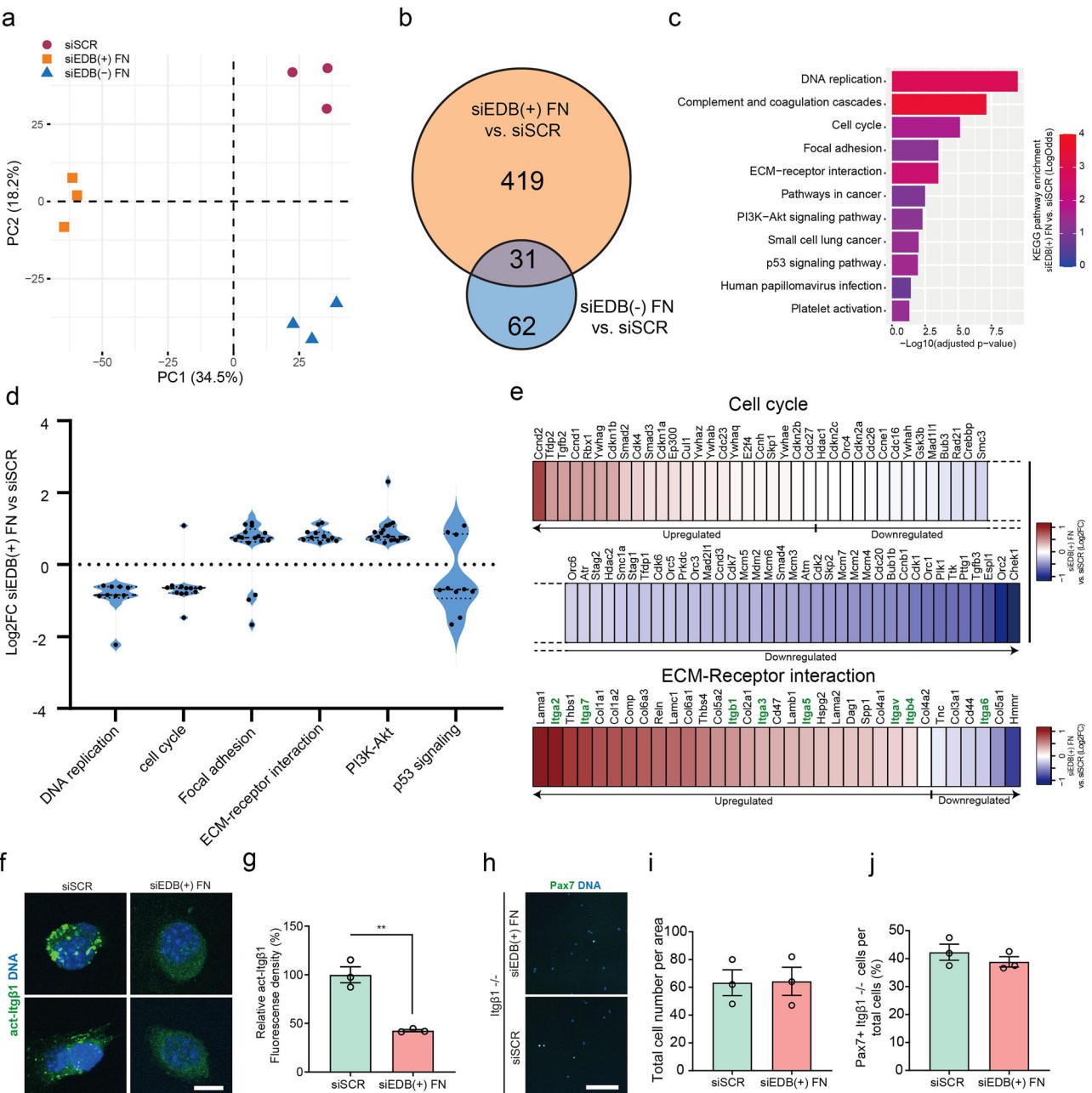

**Fig. 6 | EDB(+) FN interacts with integrins and regulates the cell cycle.**
**a** Principal component analysis (PCA) of the global proteomic signature of siEDB(+) FN and siEDB(−) FN and siSCR treated MuSC-derived myoblasts as determined by mass spectrometry. **b** Venn diagram showing the overlap of significantly regulated proteins with log2 fold change (log2FC) > 0.58 and adjusted $p$-value < 0.1 in siEDB(+) FN or siEDB(−) FN treated MuSC-derived myoblasts when compared to the siSCR condition. **c** Visualization of KEGG pathways enriched with an adjusted $p$-value < 0.05 based on differntially regulated proteins in MuSC-derived myoblasts after siEDB(+) FN treatment compared to the siSCR condition. **d** Violin plots showing the log2FC of individual proteins in selected KEGG pathways differentially affected in MuSC-derived myoblasts treated with siEDB(+) FN compared to the siSCR condition. **e** Heatmap showing log2FC of all quantified proteins in the "Cell cycle" and "ECM-Receptor interaction" KEGG pathways in MuSC-derived myoblasts treated with siEDB(+) FN compared to the siSCR condition. Integrin subunits are highlighted in green fonts. **f**, **g** Representative immunostaining and quantification

of activated integrin β1 (act-Itgβ1, green) fluorescence intensity in MuSC-derived myoblasts after treatement with siEDB(+) FN or siSCR. DNA (blue) was counterstained using DAPI. Scale bar = 10 μm. **h–j** Representative immunostaining and quantification of total cell numbers and Pax7+ (green) MuSC-derived integrin β1 knockout (Itgβ1-/-) myoblasts after treatement with siEDB(+) FN or siSCR. DNA (blue) was counterstained using DAPI. Scale bar = 200 μm. siRNAs termed siEDB(+) FN and siEDB(−) FN (**a–j**) correspond to the siRNAs with the suffix _a described in Fig. 2c. Data points represent $n$ = 3 biological replicates per condition, each from an independent myoblast line isolated from separate young mice (**a**), and bars show means from $n$ = 3 biological replicates per condition, each from an independent myoblast line isolated from separate young mice (**g**, **i**, **j**). KEGG enrichment $p$-values (**c**) were calculated using Fisher's exact test. Error bars represent SEM and $p$-values (**g**, **i**, **j**) were calculated using a one-tailed Student's $t$-test. **$p$ < 0.01. Source data are provided as a Source Data file.

## Single myofiber culture and siRNA transfection

*Extensor digitorum longus* (*EDL*) muscles were isolated by cutting the proximal and distal tendons and then digested with 0.2% collagenase (Sigma-Aldrich, C0130) in DMEM (Wisent, 319-005-CL) at 37 °C for one hour in a water bath. Single muscle fibers were dissociated mechanically using a wide-bore glass pipette, washed three times with DMEM (Wisent, 319-005-CL), and then transferred into horse serum (Wisent, 065150) coated petri dishes (Sarstedt, 83.3902). Fibers were

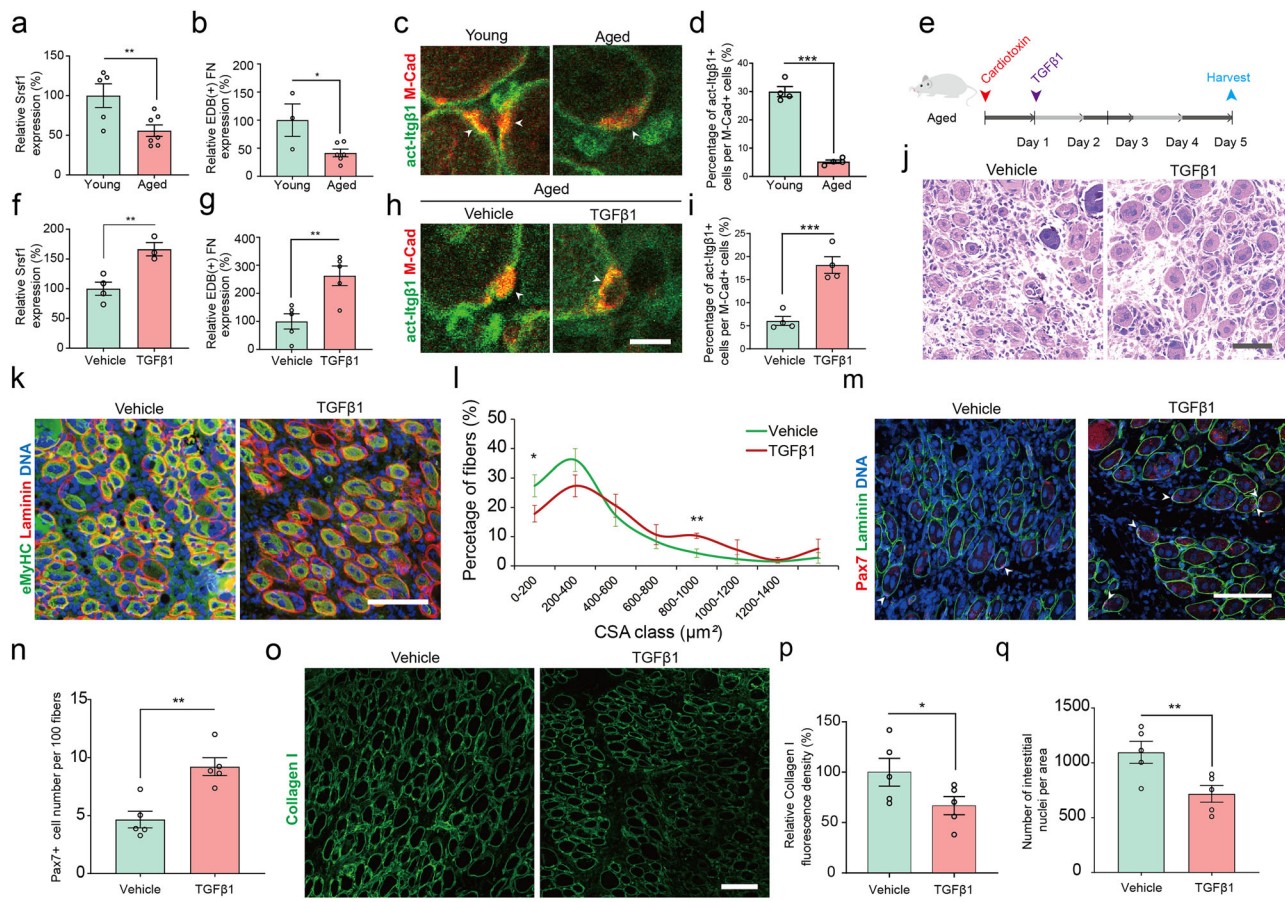

**Fig. 7 | Transient TGFβ1 treatment increases Srsf1 and EDB(+) FN levels and stimulates aged MuSCs. a**, **b** Quantitative PCR for *Srsf1* and EDB(+) FN expression comparing young and aged mouse *TA* muscles at 5 dpi. **c**, **d** Representative immunostaining and quantification of fluorescence intensity of act-Itgβ1 (green) in M-Cad (red) positive MuSCs (arrowheads) in *TA* muscle sections of young and aged mice at 5 dpi. Scale bar = 10 μm. **e** Experimental workflow for TGFβ1 treatment of regenerating aged muscles. **f**, **g** Semi-quantitative PCR analysis of *Srsf1* and EDB(+) FN in aged *gastrocnemius* muscles at 2.5 and 5 dpi respectively after TGFβ1 or vehicle treatment. **h**, **i** Representative immunostaining and quantification of fluorescence intensity of act-Itgβ1 (green) in M-Cad (red) positive MuSCs (arrowheads) in *TA* muscle sections of aged mice at 5 dpi after TGFβ1 or vehicle treatment. Scale bar = 10 μm **j** Representative H&E staining of cross sections of aged mouse *TA* muscle at 5 dpi after TGFβ1 or vehicle treatment. Scale bar = 100 μm. **k**, **l** Representative immunostaining and CSA quantification of eMyHC (green)

positive fibers in *TA* muscles of aged mice at 5 dpi after TGFβ1 or vehicle treatment. Laminin (red) and DAPI (blue) labeling DNA were used for counterstaining. Scale bar = 100 μm. **m**–**p** Representative immunostaining and quantification of Pax7+ cells (m,n, red, white arrowheads) and collagen I (**o**, **p**, green) fluorescence intensity in cross sections of aged mouse TA muscles at 5 dpi after TGFβ1 or vehicle treatment. Laminin (green) and DAPI (blue) labeling DNA were used to counterstain Pax7 (m). Scale bars = 100 μm. **q** Quantification of interstitial nuclei based on laminin and DNA staining with DAPI (as shown in m) in cross sections of aged mouse TA at 5dpi after TGFβ1 or vehicle treatment. Bars and datapoints represent means ± SEM from $n = 5$ (**a**, **g**, **n**, **p**, **q**), $n = 4$ (**d**, **f**, **i**, **l**), $n = 3$ (**b**) biological replicates, each from separate young mice, and $n = 7$ (**a**), $n = 6$ (**b**), $n = 5$ (**g**, **n**, **p**, **q**), $n = 4$ (**d**, **i**, **l**), $n = 3$ (**f**) biological replicates, each from separate aged mice. *P*-values were calculated using a one-tailed Student's *t*-test (**a**, **b**, **d**, **f**, **g**, **i**, **l**, **n**, **p**, **q**). *$p < 0.05$, **$p < 0.01$, ***$p < 0.001$. Source data are provided as a Source Data file.

maintained in growth medium containing Ham's F10 (Wisent, 318-050-CL), 20% fetal bovine serum (Wisent, 080450), 2.5 ng/mL bFGF (R&D systems, 3139-FB-025), and 1% Penicillin-Streptomycin (Wisent, 450-201-EL). For siRNA treatment, single fibers cultured for 4 h were transfected using Lipofectamine RNAiMAX (Thermo Fisher, 13778030) and opti-MEM (Life Technologies, 31985062), incubated for 72 h in an incubator at 37 °C with 5% $CO_2$, and then fixed in 2% paraformaldehyde (PFA; Thermo, 41678-5000) at room temperature.

**Immunostaining of muscle cryosections, cells, and single fibers**
10 μm thick *TA* muscle cross sections were cut with a CM1850 cryostat (Leica). For immunostaining, muscle cryosections were fixed with 4% paraformaldehyde (TCI America, P0018) or ice-cold acetone (Millipore, AX0120-8) for 10 min, permeabilized with 0.5% Triton-X-100 (Sigma-Aldrich, T8787) for 10 min, and then treated with 1% glycine (Millipore, 56-40-6) in PBS for 10 min. Sections were blocked in 5% bovine serum albumin (Thermo Fisher Scientific, BP9703100) in PBS supplemented with 0.05% Triton-X100 (Sigma-Aldrich, 1003407653)

for at least 1 h at room temperature. Primary antibodies (Supplementary Table 2) were diluted in blocking solution and incubated overnight at 4 °C in a wet chamber. Appropriate secondary antibodies (Thermo Fisher Scientific) and 4',6-Diamidino-2-Phenylindole (DAPI; Biotium, 40043) to stain DNA were applied for 1 h at room temperature. The samples were mounted using Mowiol (Sigma-Aldrich, 81381) for image acquisition. For Pax7 staining, antigen retrieval using hot 10 mM sodium citrate buffer (Sigma-Aldrich, S4641) supplemented with 0.05% Tween 20 (Bio Basic Canada, TB0560) was performed for 20 min and Fab mouse antigen fragment (Jackson ImmunoResearch, 115-007-003) was added during the blocking step. For immunostaining, cells and myofibers were fixed with 2% PFA (TCI America, P0018) in PBS, washed with PBS, treated with 1% glycine (Millipore, 56-40-6) in PBS, permeabilized with 0.05% Triton-X100 in PBS, and then blocked and processed as described above for cryosections. Fluorescence microscopy was performed with a FV1000 laser scanning confocal microscope (Olympus), and whole section imaging was performed

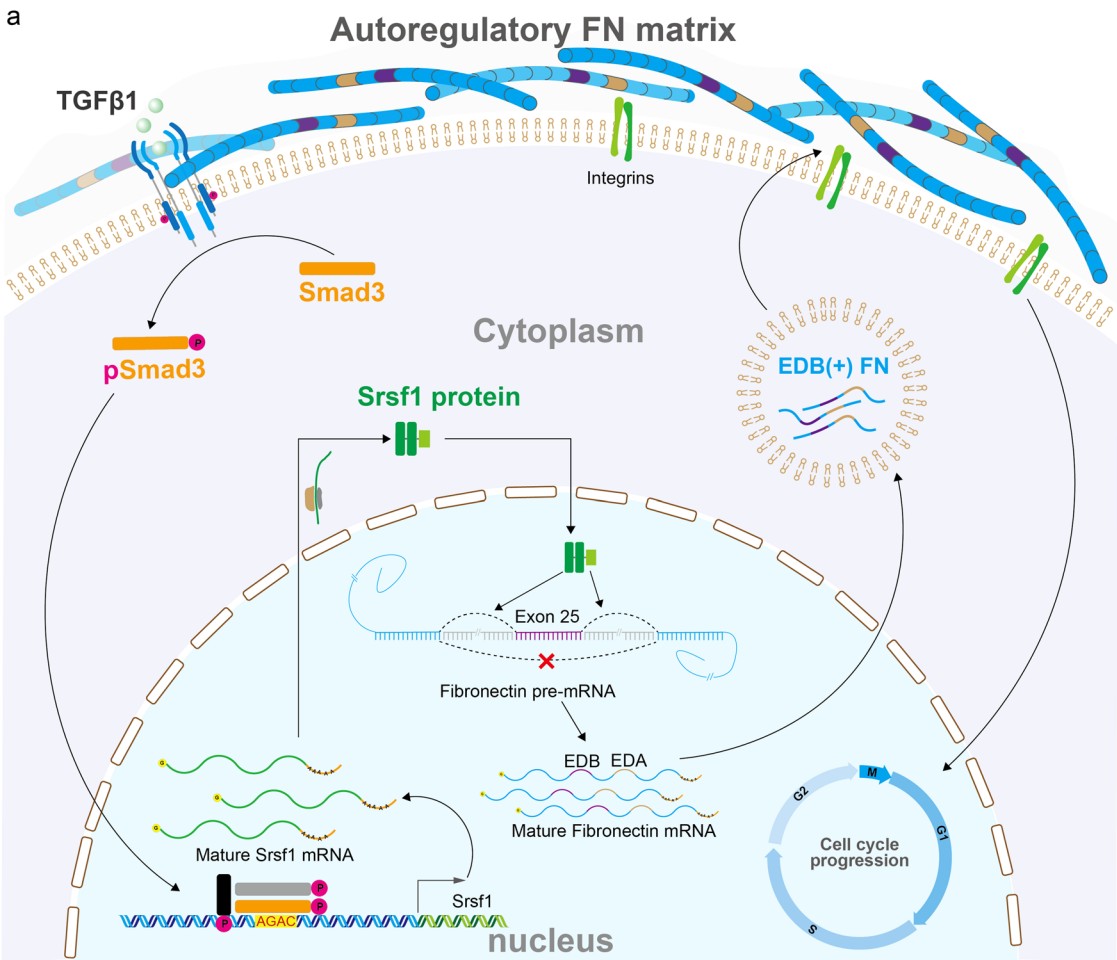

**Fig. 8 | Autoregulatory EDB(+) FN signaling in activated MuSCs. a** Schematic showing the TGFβ1-Smad3-Srsf1 pathway controlling splicing and autoregulatory EDB(+) FN secretion in activated MuSCs. EDB(+) FN, the majority of which also contains the EDA extra domain, is released into the immediate MuSC microenvironment where it activates cell-autonomous integrin signaling and stimulates cell cycle progression.

with an AxioImager M2 microscope (Zeiss). Images were quantified using ImageJ (version 1.53t)[58].

### mRNA isolation, PCR, and promoter analysis

mRNA was extracted from cells, fresh or snap frozen muscle tissues, by using TRIzol (Thermo Fisher, 15596018) according to the manufacturer's instructions. cDNA was generated using ProtoScript II Reverse Transcriptase (NEB, M0368) with 15-mers oligo-dT (Integrated DNA Technologies). For densitometric quantification of cDNA, PCR amplification was performed with a T100 Thermal Cycler (Biorad). Electrophoresis of PCR products was performed using 1.5-2% agarose gels (FroggaBio, A87-500G) in TBE buffer. Gels were stained using RedSafe Nucleic Acid Staining Solution (FroggaBio, 21141) and images were obtained using a Universal Hood gel imaging system (Bio-Rad). Grey value quantification was performed using ImageJ[58]. The percent spliced in index (PSI) was calculated using the formula: in/[in+ex] x 100 (in=exon inclusion, ex=exon exclusion). Quantitative PCR of mRNA-derived cDNA was performed using SYBR green (Meridian, BIO-98005) with an RG-6000 (Corbett) or QuantStudio 5 (Applied Biosystems) device. Primers are described in Supplementary Table 5. Transcription factor binding sites in the Srsf1 promoter were identified using Pscan[59].

### Single-cell sequencing data analysis

A normalized gene expression matrix from single-cell RNA-seq data of non-injured and injured skeletal muscle from adult 4-7 months-old C57BL/6 mice was generated based on the data by De Micheli et al.[22]

obtained from the Gene Expression Omnibus repository (GEO, GSE143437). The matrix was analyzed using Seurat Version 4.3.02[60] and clusters were identified using cell population markers. Based on the gene expression matrix, subsets of cells from uninjured and 5 dpi skeletal muscle were isolated and reanalyzed in Seurat for data scaling, identification of variable features, principal component analysis, identification of neighbors and clusters, and to calculate UMAP. FN expression density was generated with Nebulosa version 1.8.0.3[61].

### Chromatin Immunoprecipitation

For chromatin immunoprecipitation (ChIP), $10^6$ early-passage mouse myoblasts were crosslinked with 2% paraformaldehyde (TCI America, P0018) at 37 °C, washed with 1% glycine (EMD Millipore, 56-40-6) in PBS and then with ice cold PBS, and treated with SDS lysis buffer (EMD Millipore, 20-163) supplemented with protease inhibitor cocktail (EMD Millipore, 11836170001) and phosphatase inhibitor (Pierce, A32957) using the ChIP kit (EMD Millipore, 17-295) according to the manufacturer's instructions. Crosslinked chromatin was sheared on ice with a Model 120 Sonic Dismembrator (Fisher Scientific) after voltage and power optimization. Sheared chromatin was added to dilution buffer (EMD Millipore, 20-153), incubated with antibodies (Supplementary Table 2), and then loaded onto protein A magnetic beads (Thermo Fisher, 10001D). Beads were then washed with low salt buffer (EMD Millipore, 20-154), high salt buffer (EMD Millipore, 20-155), LiCl buffer (EMD Millipore, 20-156) and TE buffer (EMD Millipore, 20-157). The immunoprecipitated protein-DNA complex was then eluted from

beads with freshly made elution buffer containing 1% SDS (Spectrum,155-21-3), and 0.1 M NaHCO₃ (Sigma-Aldrich,144-55-8). Cross-linking was then reversed by incubating the samples with 5 M NaCl (EMD Millipore, 20-159) at 65 °C for 4 h. Samples were then digested with Proteinase K (Bio basic, PB0451), DNA was purified using phenol/chloroform (Thermo Fisher, 15593031), pelleted DNA was resuspended in ddH₂O and diluted for PCR analysis with primers (Supplementary Table 5) designed to flank the Smad3 and Smad5 motifs.

## Molecular modeling

In order to visualize interactions between the RRM domain of Srsf1 and the GA-rich RNA sequence in the FN EDB exon, molecular docking was performed using the MOE software (Chemical Computing Group, version MOE 2024.06). Because the RRM domain is known to bind to a GA-rich pre-mRNA motif, we relied on the structure of human Srsf1 (PDB ID: 2M8D)[26], which has 100% homology with the mouse sequence, in complex with the RNA motif 5′-UGAAGGAC-3′. Firstly, the NMR structure was prepared using the MOE software QuickPrep tool, which includes adding charges, protonation, solvent addition, and bond fixation. The AMBER10: EHT force field was used. Subsequently, starting from the RNA sequence 5′-UGAAGGAC-3′ of the complex, four mutations were made using the MOE software sequence editor to obtain the new RNA sequence 5′-AGGAGAAG-3′. Energy minimization of the complex was performed, followed by a molecular dynamics' simulation for 0.1 seconds at 300 K and 100 kPa. Finally, a second energy minimization was performed.

## Mass spectrometry

Primary mouse myoblasts were solubilized in 8 M urea (Fisher Chemicals, U15-500) lysis buffer supplemented with 1 M ammonium bicarbonate (Fisher Scientific, A643-500), 20 mM HEPES (Wisent, 330-050-el), and were sonicated on ice with a Model 120 Sonic Dismembrator (Fisher Scientific). After the solution was adjusted to a final urea concentration of 3 M, protein was quantified by BCA (Bio Basic, SK3021). 50 µg protein was treated with 5 mM dithiothreitol (DTT, Thermo Fisher, R0861) at 95 °C for 2 minutes, and incubated at RT for 30 min. Samples were then treated with 7.5 mM chloroacetamide (Sigma-Aldrich, C0267) in the dark at RT for 20 min, diluted with 1 M ammonium bicarbonate to a final concentration of 2 M urea and digested with 1 µg MS-grade trypsin (Pierce, 90058) at 30 °C overnight. Samples were then acidified by adding MS-grade trifluoroacetic acid (TFA, Fisher Scientific, A116-50) to a final concentration of 0.2%, followed by Ziptip (Pierce, 87784) clean up according to the manufacturer's instructions. HPLC-grade water (Sigma-Aldrich, 270733) was used for all reagents and buffers. Cleaned peptides (500 ng) were injected into an HPLC (nanoElute, Bruker Daltonics) equipped with a trap column (Acclaim PepMap100 C18, 0.3 mm id x 5 mm Dionex Corporation) and an analytical C18 column (1.9 µm beads, 75 µm x 25 cm, PepSep). Peptides were separated for 2 h using a linear gradient of 5-37% acetonitrile in 0.1% formic acid at a flow rate of 400 nl/min while being injected into a TimsTOF Pro ion mobility mass spectrometer equipped with a Captive Spray nano electrospray source (Bruker Daltonics). The target intensity was set to 20,000, with an intensity threshold of 2500. Sample acquisition was performed in diaPASEF mode. Briefly, for each single TIMS (100 ms) in diaPASEF mode, 1 mobility window consisting of 27 mass steps (m/z between 114 and 1414 with a mass width of 50 Da) was used per cycle (1.27 seconds duty cycle) with collision energy of 42.0 eV. These steps cover the diagonal scan line for +2 and +3 charged peptides in the m/z-ion mobility plane. Raw data was analyzed using DIA-NN software (version 1.9.2)[62] and the Uniprot[63] mouse proteome database dated 07/03/2021 with 55,366 entries. The unique genes matrix output from DIA-NN was used to calculate fold changes and adjusted p-values with the DIA-Analyst suite from Monash University. Significantly affected proteins were defined as having adjusted p < 0.1 and an absolute log2 Fold change >0.58.

Kegg pathway analysis was performed in the DIA-Analyst suite. Results were exported as tables, and data were visualized using R Studio 2022.07.2 + 576. Mass spectrometry proteomics raw files and the respective data analysis files have been deposited to the ProteomeXchange Consortium (http://proteomecentral.proteomexchange.org) via the PRIDE partner repository with identifier PXD062037.

## Statistics and Reproducibility

All data points, and the n values reported in the figure legends, correspond to biological replicates representing independent experimental repeats, each derived from cells or tissues of separate mice or human donors. Representative images were confirmed in at least 3 independent experiments with similar results. Experiments were designed to include male and female mice in approximately equal proportions, and animals were randomized into experimental groups. Data was disaggregated for sex, but no differences were observed in any of the experiments. Therefore, samples/datapoints were pooled and considered equal biological replicates. Sample size determination was based on the expected effect size and variability that was previously observed for similar readouts in the investigator's laboratory. In vivo treatments were not blinded, but image analysis was performed in a blinded manner. Statistical analysis was performed using GraphPad Prism (GraphPad Software, version 9.4.1 (681)). Statistical significance for binary comparisons was assessed by a student's t-test after verification that variances didn't differ between groups or by a Welch correction when variance was observed. For comparison of more than two groups, one-way ANOVAs were used and followed by Dunnett's post-hoc testing for comparing different means to the control group, or Tukey's post-hoc testing for comparing all means. All data are expressed as mean ± SEM.

## Reporting summary

Further information on research design is available in the Nature Portfolio Reporting Summary linked to this article.

## Data availability

No new code was generated in the context of this publication. The single-cell sequencing dataset analysed in this study is available in Gene Expression Omnibus (GEO, https://www.ncbi.nlm.nih.gov/geo/) under the accession number GSE143437. Mass spectrometry proteomics raw files and the respective data analysis files have been deposited to the ProteomeXchange Consortium (http://proteomecentral.proteomexchange.org) via the PRIDE partner repository with identifier PXD062037. The structure of human Srsf1 is available in the RCSB Protein Data Bank (https://www.rcsb.org/) under structure number 2M8D. Source data are provided with this paper.

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

## Acknowledgements

The authors acknowledge support by the Quebec Network of Research on Ageing, which provided access to their colony of aged mice. We thank Honglin Chen and Richard Hynes, respectively, for making the Srsf1 cDNA and the FN minigene plasmid containing the EDB exon and flanking sequences available through Addgene[27,64]. We are grateful to the Cosgrove laboratory for making their single-cell sequencing data, which we reanalyzed in this study, available on Gene Expression Omnibus (GEO, GSE143437)[22]. We acknowledge the availability of the published human Srsf1 structure (PDB ID: 2M8D), which served as the basis for the molecular docking analyses conducted in this study[26]. We thank Dominique Lévesque and the Plateforme de protéomique par spectrométrie de masse at the Université de Sherbrooke for support with mass spectrometry sample processing and raw data acquisition. C.F.B. was supported by the Canadian Institutes of Health Research (CIHR, PJT-162442 and PJT-507167), the Natural Sciences and Engineering Research Council of Canada (NSERC, RGPIN-2017-05490 and RTI-2025-00383), the Fonds de Recherche du Québec - Santé (FRQS, Dossiers 296357, 34813, and 36789), the Fonds de Recherche du Québec - Nature et Technologies (FRQNT, Dossier 331297), the ThéCell Network (supported by the FRQS), the Canadian Stem Cell Network, and a research chair of the Centre de Recherche Médicale de l'Université de Sherbrooke (CRMUS). P.L.B. was supported by NSERC (RGPIN-2022-04028 and RTI-2020-00560) and CIHR (PJT-480163). Y.L. was supported by a post-doctoral fellowship by the Muscular Dystrophy Canada Research Fellowship in collaboration with the Neuromuscular Disease Network for Canada. S.D. was supported by a PhD fellowship by the FRQS (Dossier 305231). S.C.S. was supported by a postdoctoral fellowship by the German Research Foundation (DFG, 505064275).

## Author contributions

C.F.B. and Y.L. initiated and managed the project. Y.L. designed and conducted the majority of experiments. S.C.S. contributed to experiments with naturally aged mice and performed proteomic analysis. S.D. provided bioinformatic support and P.L. performed molecular modeling. F.B. and S.B provided human tissues for cell isolation. T.D. and J.N.F. contributed to experiments with human myoblasts. C.F.B. and Y.L. interpreted results and wrote the manuscript, which was reviewed and edited by all authors.

## Competing interests

C.F.B. received consultancy fees from BioAge Labs, Richmond, California, but these had no relation to the content of this publication. J.N.F., T.D. and C.F.B. are or were employees of the Société des Produits Nestlé SA, Switzerland. The other authors declare no competing interests.
