## [Transparent Peer Review file · Nature Communications]

TGF β -Smad3 Signaling Restores Cell-Autonomous Srsf1-Mediated Splicing of Fibronectin in Aged Skeletal Muscle Stem Cells

Corresponding Author: Professor C. Florian Bentzinger

Version 0:

Reviewer comments:

Reviewer #1

(Remarks to the Author)

The manuscript addresses an important aspect of skeletal muscle regeneration and aging, focusing on the role of fibronectin (FN) and its splicing in muscle stem cells (MuSCs). Understanding this mechanism could lead to novel therapeutic strategies for age-associated regenerative failure. The identification of the TGF β -Smad3-Srsf1-EDB(+) FN axis as a critical pathway in MuSC function and its decline with aging is novel. The study employs a variety of techniques, including single-cell sequencing, immunostaining, siRNA knockdown, chromatin immunoprecipitation, and in vivo mouse models. This multifaceted approach strengthens the validity of the findings. The figures and extended data are well-organized and provide a clear visual representation of the experimental results, supporting the conclusions drawn by the authors.

While the study establishes the importance of the TGF β -Smad3-Srsf1-EDB(+) FN axis, the exact molecular mechanisms by which EDB(+) FN specifically regulates MuSC proliferation and function are not fully elucidated. Further mechanistic insights would strengthen the manuscript. The study is based on mouse models, and while the findings are significant, it remains to be seen how well these results translate to human aging and muscle regeneration. Some discussion or preliminary data on human MuSCs would enhance the relevance of the study. Finally, the manuscript should address potential off-target effects of the siRNA treatments and TGF β 1 injections. Additional controls or experiments to rule out these effects would make the conclusions more robust.

Suggested Experiments to Strengthen Conclusions:

1. Mechanistic Studies on EDB(+) FN Function: Conduct experiments to delineate the specific signaling pathways and molecular interactions mediated by EDB(+) FN in MuSCs. For example, proteomic analysis of MuSCs treated with EDB(+) FN could identify interacting partners and downstream effectors.
2. Human MuSC Validation: Perform key experiments using human MuSCs to validate whether the TGF β -Smad3-Srsf1-EDB(+) FN axis operates similarly in humans. This could involve analyzing FN splicing and Srsf1 expression in human MuSCs from young and aged donors.
3. Long-Term Effects of TGF β 1 Treatment: Investigate the long-term effects of transient TGF β 1 treatment on muscle regeneration and fibrosis in aged mice. This would provide insights into the potential therapeutic applications and safety of this approach.
4. Off-Target Effects and Specificity: Include additional controls to rule out off-target effects of siRNA and TGF β 1 treatment. For example, use scrambled siRNA and a different growth factor that does not activate Smad3 to demonstrate specificity.

Reviewer #2

(Remarks to the Author)

Liu Y. et al. aim to understand the underlying mechanisms of sarcopenia, a clinically significant area of research. They investigated the role of cellular fibronectin-specific isoforms during myogenesis. The manuscript is well-written and structured. Despite the amount of work contributed to this paper, there are the following methodological and statistical concerns:

1. In the 'Materials and Methods' section, the authors initially describe the study population as young and aged mice. However, in most figures, starting with Figure 1, the authors refer to the mice as adult. This discrepancy in the age of the mice used in the study needs to be clarified to ensure the accuracy of the research. Please provide a clear characterization

of the mouse study group (young versus aged).

2. In the figure legend, the authors characterize the samples as “independent biological replicates per condition.” Specifying the number of individual animals collected from the samples would help ascertain the presented data. Independent biological replicate is an extensive term that can be applied to individual cells isolated from the same animal. Furthermore, the study would greatly benefit from increased sample sizes, demonstrating data reproducibility under the same conditions in several independent experiments.
3. It is not clear why the authors dismiss the role of EDA(+)FN in MuSC differentiation. While their data does demonstrate that the levels of EDB(+) are more prominent in myoblasts, compared to FAPs, it's important to understand the reason for dismissing the role of EDA(+). Furthermore, to support the authors' hypothesis, it would be beneficial to quantify the overlapping regions of EDB(+) and EDA(+) FN with the corresponding markers using several animals and several fields of view, including images from the extended data (for example, Ext. Data Fig 1h where colocalization of both isoforms with f4/80 looks comparable).
4. The authors mention that the experiment with the transfer of mouse fibroblasts with isoform-specific coding plasmids revealed an enrichment of the fibroblast ECM but did not quantify that enrichment following the transformation. Also, it would be beneficial to support the presented data by repeating the cell culture using EDA(+) and EDB(+) coated coverslips using an unaltered mouse fibroblast cell line.
5. While investigating the mechanism of EDB(+)FN in myogenesis, the authors mention the correlation of p-Smad3 with Srsf1 but do not present correlation figures or a statistical approach used to quantify the results. Considering that in Fig. 5h-i, there is no significant difference between EDB(+) and vehicle (i), but Srsf1 expression is significant (h, vs. vehicle), it would also be interesting to see the correlation results of EDB(+)FN and Srsf1 expressions. What is the minimum expression of Srsf1 that would support EDB(+)FN expression?
6. It has been previously published that silencing Srsf1 reduces the inclusion of EDA isoform in fibronectin. Although the authors present the data that singles out the role of the EDB isoform, it would be interesting to see the expression of both isoforms upon SIS3 treatment.
7. It is crucial to ascertain the expression of both isoforms to confirm the continuous primary role of EDB(+)FN in MuSC differentiation. This comprehensive understanding is essential for advancing the field of research on sarcopenia.
8. The manuscript would be significantly enhanced by the inclusion of EDB-knockout mice, particularly aged mice. This addition would provide a more comprehensive understanding of the role of cellular fibronectin-specific isoforms during myogenesis.
9. The paper would be more comprehensive with the inclusion of a limitation section in the discussion. This would help the authors acknowledge potential shortcomings and provide a more balanced view of their research.

Reviewer #3

(Remarks to the Author)

Version 1:

Reviewer comments:

Reviewer #1

(Remarks to the Author)

I would like to thank the authors for their comprehensive and thoughtful revision. The additional data, methodological clarifications, and expanded discussion—particularly regarding the role of EDA(+) fibronectin and the inclusion of study limitations—have greatly improved the manuscript. The study provides novel insights into the TGF β –Smad3–Srsf1–EDB(+) FN signaling axis and its relevance in MuSC regulation and sarcopenia, and I believe it will be of strong interest to the field. Congratulations on this valuable contribution.

Reviewer #2

(Remarks to the Author)

The reviewer appreciates the authors' responses, their implementation of the suggestions, and their attention to the comments in the revised version. However, several concerns remain that require additional attention.

1. The authors state in their reply “now also show that almost all EDB(+) FN is in fact EDA(+) (Supplementary Fig. 1c-e)”; therefore, how can they be certain that the observed effect is actually due to EDB(+) and not because the “transcripts containing EDB also contain the EDA domain in myoblasts”?
2. Figure 5 legend indicates, “myoblasts isolated from n=3 young mice per condition”, while the Figure 6a description states, “Principal component analysis (PCA) revealed that biological replicates from all three conditions showed distinct clustering (Fig. 6a)” and the legend describes “Data points and bars represent means from myoblasts isolated from n=3 (a-j) young mice per condition”. Figure 6a has only 3 data points. Therefore, both statements contradict each other. Altogether, the answer to the second comment remains unclear. Are there replicates or single animals? Is a single data point being presented as an average of 3 replicates from a single animal? Please clarify the following in the Statistical analysis, the main text, and the figure legends. Furthermore, if you have used 3 individual animals and had biological replicates, why cannot the sample size be increased by presenting all replicates of an individual animal or human used in the analysis?

3. Please provide the representative images for Supplementary Fig. 1j,k.
4. The authors haven't presented any correlation analysis. Therefore, the authors should refrain from using the term "correlation" and replace it with "associated".
5. In their reply, the authors state that "the plasmids containing EDB(-)EDA(+) FN, pFN, or the empty vector, showed no difference", but their figure demonstrates that EDB(-)EDA(+)FN is statistically different from the empty vector. Although the following statement is not written in the manuscript, this difference might play an important role in myogenesis.
6. The Limitation section should include the small sample size and the absence of a sex difference investigation (using both male and female mouse samples).
7. What is the clinical significance and future directions of this study?

Reviewer #3

(Remarks to the Author)

Version 2:

Reviewer comments:

Reviewer #2

(Remarks to the Author)

There are no further questions.

Reviewer #3

(Remarks to the Author)

REVIEWER COMMENTS

We thank our reviewers for their helpful and constructive feedback. As detailed in our point-by-point response below, we revised our manuscript and addressed our reviewers' comments with extensive additional experiments and textual revisions. Briefly, (I) we performed a proteomic analysis to study EDB(+) FN downstream signaling and interaction partners, which revealed an involvement of integrins and cell cycle regulators (**Fig. 6a-j, Fig. 7c,d,h,i, and Supplementary Fig. 6a,b**). (II) We validated that the TGF13-Smad3-Srsf1-EDB(+) FN axis is active in human myogenic progenitors (**Supplementary Fig. 3c-f, 4g-i, 5c-f**) and (III) performed additional control experiments including testing of a growth factor unrelated to TGF131 (**Supplementary Fig. 7a-c**). (IV) We now also provide several additional experiments focused on the EDA extra domain, including quantification of its tissue distribution (**Supplementary Fig. 1j,k**) and an experiment to analyze its relative amount of splicing compared to EDB (**Supplementary Fig. 1c-e**). (V) We quantified enrichment of EDB(+)EDA(-) and EDB(-)EDA(+) FN in the ECM of plasmid-transfected fibroblasts before co-culture with myoblasts (**Supplementary Fig. 2a-c**) and (VI) decellularized the fibroblast cultures before seeding myogenic cells onto the substrate to exclude interference by cell-cell contacts or soluble signals (**Supplementary Fig. 2d-f**). (VII) We now also demonstrate that our timed single TGF131 injection protocol has no adverse long-term effects on skeletal muscle regeneration and fibrosis (**Supplementary Fig. 8a-g**). Lastly, (VIII) we performed additional experiments to demonstrate the temporal dynamics and kinetics of Smad3 inhibition with respect to its effects on Srsf1 and EDB(+) FN expression (**Supplementary Fig. 5a-d**) and (IX) added a paragraph about limitations of the study to the discussion section of our manuscript (**page 15, paragraph 3**). Altogether, we believe that these new data considerably support and strengthen our original conclusions, and the edits further enhance our manuscript. We hope that it is now considered suitable for publication in Nature Communications.

Reviewer #1 (Remarks to the Author):

The manuscript addresses an important aspect of skeletal muscle regeneration and aging, focusing on the role of fibronectin (FN) and its splicing in muscle stem cells (MuSCs). Understanding this mechanism could lead to novel therapeutic strategies for age-associated regenerative failure. The identification of the TGF13-Smad3-Srsf1-EDB(+) FN axis as a critical pathway in MuSC function and its decline with aging is novel. The study employs a variety of techniques, including single-cell sequencing, immunostaining, siRNA knockdown, chromatin immunoprecipitation, and in vivo mouse models. This multifaceted approach strengthens the validity of the findings. The figures and extended data are well-organized and provide a clear visual representation of the experimental results, supporting the conclusions drawn by the authors.

We appreciate the reviewer's positive feedback and are grateful for their time and constructive criticism.

While the study establishes the importance of the TGF13-Smad3-Srsf1-EDB(+) FN axis, the exact molecular mechanisms by which EDB(+) FN specifically regulates MuSC proliferation and function are not fully elucidated. Further mechanistic insights would strengthen the manuscript. The study is based on mouse models, and while the findings are significant, it remains to be seen how well these results translate to human aging and muscle regeneration. Some discussion or preliminary data on human MuSCs would enhance the relevance of the study. Finally, the manuscript should address potential off-target effects of the siRNA treatments and TGF131 injections. Additional controls or experiments to rule out these effects would make the conclusions more robust. Suggested Experiments to Strengthen Conclusions:

1. Mechanistic Studies on Function: Conduct experiments to delineate the specific signaling pathways and molecular interactions mediated by EDB(+) FN in MuSCs. For example, proteomic analysis of MuSCs treated with EDB(+) FN could identify interacting partners and downstream effectors.

We thank the reviewer for this suggestion. We fully agree that additional insights into EDB(+) FN interacting partners and downstream effectors would strengthen our paper and complement our discovery of the intricate upstream mechanisms involving alternative splicing through the TGF β -Smad3-Srsf1 axis. As described in our present manuscript we discovered that activated myogenic progenitors express high levels of cell-autonomous EDB(+) FN, which in itself will bind to target receptors and induce downstream signaling. Therefore, exposure to additional exogenous EDB(+) FN protein likely only leads to potentiation of receptors and downstream pathways that are already in an active state. Based on these considerations, we reasoned that proteomic profiling of an EDB(+) FN loss-of-function paradigm should allow us to observe the largest magnitude of changes in downstream signaling when compared to control cells. To this end, we performed the suggested proteomics experiment by comparing the effects of siEDB(+) FN, siEDB(-) FN, and scrambled siRNA (siSCR) in mouse myoblasts.

Principal component analysis (PCA) of the resulting proteomic profiles showed distinct and reproducible clustering of biological replicates for siEDB(+) FN, siEDB(-) FN, and siSCR (**Fig. 6a**). In agreement with the pronounced reduction in myogenic progenitor numbers *in-vitro* and *in-vivo* after siEDB(+) FN treatment (**Fig. 2f,g and 3f,g**), we observed that the KEGG pathways “DNA replication” and “Cell cycle” were significantly downregulated in knockdown cells when compared to the siSCR or siEDB(-) FN condition (**Fig. 6c-e**). Suggesting a compensatory response, siEDB(+) FN induced a notable upregulation of genes involved in the KEGG pathways “ECM-receptor interactions” and “Focal adhesions” including increased levels of integrin subunits and integrin-binding ECM components such as laminin, collagen, and matricellular thrombospondin (**Fig. 6e**).

To further investigate a potential involvement of integrins we used an antibody detecting the active conformation of integrin beta-1 (Itg β 1), the most expressed integrin subunit in MuSCs (Schüler et al. Front Cell Dev Biol. 2022). This experiment revealed that levels of active Itg β 1 in myoblasts are dramatically reduced upon siEDB(+) FN treatment compared to siSCR (**Fig. 6f,g**). Moreover, Itg β 1 knockout (Itg β 1 KO) myoblasts show proliferation defects resembling cells treated with siEDB(+) FN (Rozo et al. Nature Medicine 2016). Thus, we decided to assess whether the phenotype of Itg β 1 KO cells can be further aggravated by loss of cell-autonomous EDB(+) FN. While siEDB(+) FN led to a 71-80% reduction in cell numbers in wt cells (**Fig. 2g**), we observed no statistically significant effect in Itg β 1 KO cells compared to siSCR (**Fig. 6h-j**). Lastly, in agreement with published work (Rozo et al. Nature Medicine 2016), we also observed that aged MuSCs contain lower levels of activated Itg β 1 compared to young cells (**Fig. 7c,d**), and stimulation of EDB(+) FN expression using our transient TGF β 1 treatment protocol was able to partially reverse this phenotype (**Fig. 7h,i**).

Altogether, these results support the notion that cell-autonomous EDB(+) FN in myogenic progenitors interacts with an integrins and controls downstream signaling pathways involved in cell cycle progression (**Fig. 8a**).

2. Human MuSC Validation: Perform key experiments using human MuSCs to validate whether the TGF β -Smad3-Srsf1-EDB(+) FN axis operates similarly in humans. This could involve analyzing FN splicing and Srsf1 expression in human MuSCs from young and aged donors.

We appreciate the reviewers feedback and acknowledge the importance of extrapolation to human biology. We observed that that FN is only expressed by activated MuSCs. Thus, freshly isolated

cells from healthy human tissue, which would be largely quiescent, would not be suitable for assessing TGF13-Smad3-Srsf1-EDB(+) FN signaling. To circumvent this problem, we resorted to isolated human myoblasts at the earliest possible passage under proliferative conditions in culture. Our team collaborates with an orthopedic surgeon producing skeletal muscle tissue as a byproduct of knee remodeling procedures. Because most patients getting these procedures are in their 20s or 30s, the timespan of this revision didn't allow us to obtain sufficient biological myoblast replicates from aged patients. We also ruled out the possibility to source aged myoblasts from commercial suppliers as they are typically sold at least at passage four leading to culture adapted gene expression profiles.

Thus, to validate that the TGF13-Smad3-Srsf1-EDB(+) FN axis operates similarly in adult human myoblasts we first performed knockdown experiments using two different siRNAs each for EDB(+) and EDB(-) FN, as well as the siSCR control. Resembling its effects in mouse cells, siEDB(+) led to a 55-58% reduction in the number of human myoblasts while siEDB(-) had no negative impact compared to siSCR (**Supplementary Fig. 3c-f**). Furthermore, in human myoblasts, two different siRNAs targeting Srsf1 reduced EDB(+) FN expression by 82-83% compared to the siSCR condition (**Supplementary Fig. 4g-i**). Lastly, the Smad3 inhibitor SIS3 decreased the expression of Srsf1 by 72% and EDB(+) FN by 90%, while reducing human myoblast numbers by 87% compared to the vehicle control (**Supplementary Fig. 5c-f**). In light of the well established role of TGF131 in activating Smad3 in mammalian cells, these results support the notion that the function of the TGF13-Smad3-Srsf1-EDB(+) FN axis is conserved from mice to humans.

3. Long-Term Effects of TGF131 Treatment: Investigate the long-term effects of transient TGF131 treatment on muscle regeneration and fibrosis in aged mice. This would provide insights into the potential therapeutic applications and safety of this approach.

We thank the reviewer for raising this important point. We initially assessed muscle regeneration following single-dose TGF131 treatment in aged mice at 5 days post injury (dpi) (**Fig. 7e-q**). Our data revealed that TGF131 stimulation increased EDB(+) FN levels and Itg131 activation resulting in improved muscle regeneration with more Pax7+ cells (**Fig. 7j-n**). Importantly, we also observed that the treatment resulted in a lower number of interstitial cells, which is indicative of reduced fibrosis (**Fig. 7q**). We now provide a second fibrosis readout at 5 dpi showing that collagen I deposition is decreased by 33% in the TGF131 group compared to the vehicle control (**Fig. 7o,p**).

To address the reviewer's question, we treated skeletal muscle of aged mice with the same single-dose TGF131 protocol and analyzed the tissue at 30 days post injury (30 dpi) (**Supplementary Fig. 8a**). Compared to 5 dpi, skeletal muscle tissue at 30 dpi contains little interstitial volume and fibers are more mature and tightly packed (**Supplementary Fig. 8b**). Therefore, we were not able to reliably quantify cell numbers in the interstitial space. As an alternative fibrosis marker, we analyzed collagen I deposition and observed that it was not significantly different between the vehicle and TGF131 condition (**Supplementary Fig. 8d,e**). We also did not observe any negative effects on fiber size or the number of Pax7+ cells (**Supplementary Fig. 8c and 8f,g**).

From these observations we conclude that single-dose TGF131 treatment of aged muscles stimulates MuSC function, improves tissue regeneration, and reduces fibrosis in the acute phase after injury without causing negative long-term effects.

4. Off-Target Effects and Specificity: Include additional controls to rule out off-target effects of siRNA and TGF131 treatment. For example, use scrambled siRNA and a different growth factor that does not activate Smad3 to demonstrate specificity.

These are excellent suggestions and we have thoroughly implemented them in our manuscript. To ensure that siRNAs have no off-target effects, we included a non-coding scrambled siRNA (siSCR) in all experiments. For most experiments we also used two different experimental siRNAs for each target. Importantly, we observed specific effects in the two siEDB(+) conditions that were neither present in the siSCR control nor in the two siRNAs targeting the EDB(-) form of FN. Thus, by considering both the siSCR and siEDB(-) conditions as controls, most siRNA experiments were in fact double or triple controlled. Wherever possible, we also backed up results using alternative readouts (e.g. gel densitometry & quantitative PCR for mRNA readouts, or interstitial cell numbers & collagen I staining for fibrosis readouts). Lastly, for drug treatments we always included a vehicle control. Based on the reviewer's suggestion, we also tested epidermal growth factor (EGF) as a growth factor unrelated to TGF β 1. In contrast to TGF β 1, EGF did not activate Smad3 or induce EDB(+) FN splicing (**Supplementary Fig. 7a-c**).

Reviewer #2 (Remarks to the Author):

Liu Y. et al. aim to understand the underlying mechanisms of sarcopenia, a clinically significant area of research. They investigated the role of cellular fibronectin-specific isoforms during myogenesis. The manuscript is well-written and structured. Despite the amount of work contributed to this paper, there are the following methodological and statistical concerns:

We appreciate the reviewer's positive assessment of our manuscript and are grateful for their time and constructive criticism.

1. In the 'Materials and Methods' section, the authors initially describe the study population as young and aged mice. However, in most figures, starting with Figure 1, the authors refer to the mice as adult. This discrepancy in the age of the mice used in the study needs to be clarified to ensure the accuracy of the research. Please provide a clear characterization of the mouse study group (young versus aged).

We thank the reviewer for bringing this to our attention. We now specify these details in the methods section of our manuscript on **page 20, paragraph 1** (header "Mice"). Throughout our study we exclusively used "young" mice that were between 6-8 weeks-old and "aged" animals that were between 22-25 months-old. The only data from 4-7 months-old "adult" mice in our manuscript originated from the publicly available single-cell sequencing dataset we reanalyzed for **Fig. 1a** (De Micheli et al., Cell Rep., 2020).

2. In the figure legend, the authors characterize the samples as "independent biological replicates per condition." Specifying the number of individual animals collected from the samples would help ascertain the presented data. Independent biological replicate is an extensive term that can be applied to individual cells isolated from the same animal.

We appreciate this suggestion and have revised the manuscript text for more clarity. We now detail in the methods section on **page 28, paragraph 1** (header "Statistical analysis") that all datapoints throughout all experiments were obtained from biological replicates each originating from an individual mouse or human. (i.e. each dot in the scatter clouds overlaid with graphs in our figures represents samples or cells from an individual animal or human). To prevent ambiguity, we now also specify this more clearly in the figure legends where N numbers are provided for each experiment.

Given that our study depends largely on tissues and primary cells derived from mice, we carefully implemented our institutional and funding guidelines regarding the 3R directives for

reduction of animal experimentation. Thus, in order to determine sample size we always use a power calculation based on the expected effect size and variability that was previously observed for similar readouts in our laboratory. For instance, we observed a certain intrinsic variability in experimental cohorts with aged mice and had to use higher numbers of animals for some readouts (**Fig. 7a-q**). Fortunately, across this study, effect sizes were often substantial, which allowed us, even with stringent testing parameters, to detect statistical differences with comparably modest sample sizes.

We would also like to highlight that we included an extensive set of control experiments and alternative complementary readouts to support our conclusions. This includes the use of two different experimental siRNAs for most targets, as well as a non-coding scrambled siRNA (siSCR) control. Notably, we observed specific effects in the two siEDB(+) conditions that were neither present in the siSCR control nor in the two siRNAs targeting the EDB(-) form of FN. Thus, most siRNA experiments were in fact double or triple controlled. Wherever possible, we also backed up results using alternative readouts (e.g. gel densitometry & quantitative PCR for mRNA readouts, or interstitial cell numbers & collagen I staining for fibrosis readouts). Moreover, for drug treatments we always included a vehicle control and, as suggested by reviewer #1, we also tested EGF as a growth factor unrelated to TGF β 1 to demonstrate pathway specificity (**Supplementary Fig. 7a-c**).

3. It is not clear why the authors dismiss the role of EDA(+)/FN in MuSC differentiation. While their data does demonstrate that the levels of EDB(+) are more prominent in myoblasts, compared to FAPs, it's important to understand the reason for dismissing the role of EDA(+). Furthermore, to support the authors' hypothesis, it would be beneficial to quantify the overlapping regions of EDB(+) and EDA(+)/FN with the corresponding markers using several animals and several fields of view, including images from the Supplementary (for example, Ext. Data Fig 1h where colocalization of both isoforms with f4/80 looks comparable).

This is an excellent point, and we carefully revised our manuscript to make sure not to dismiss a potential role of EDA. Our reasoning for focussing the experiments on the EDB domain was largely due to its remarkable specificity as an ECM marker for activated MuSCs, while EDA showed a much broader tissue distribution (**Fig. 1e-h**). Moreover, we only observed a pro-proliferative effect for EDB(+)/EDA(-) FN, but not for EDB(-)/EDA(+) FN when overexpressed in fibroblasts and co-cultured with myoblasts (**Fig. 2a,b**). As outlined below, we repeated this experiment in the absence of living fibroblasts and were able to confirm that the EDB domain alone is sufficient for the pro-myogenic effects on myoblasts (**Supplementary Fig. 2d-f**).

As suggested by the reviewer we now also provide a quantification of the area of overlap of the EDB- and EDA-specific staining with MuSCs in skeletal muscle sections from multiple animals for each condition. This data confirmed that EDB(+) FN was largely restricted to the immediate microenvironment of activated MuSCs (91% co-localization with MuSCs) while EDA(+) FN was broadly distributed in the tissue (14% co-localization with MuSCs) (**Supplementary Fig. 1j,k**). In addition, we performed a new PCR experiment in which we used mRNA from proliferating mouse myoblasts with primers spanning a single amplicon containing both the EDB and EDA extra exons. This experiment revealed that almost all mRNA containing EDB, also contains EDA (**Supplementary Fig. 1c-e**). In contrast, expression of FN mRNA species containing only either EDB or EDA, or double negative transcript (pFN), appear to be less abundant in myoblasts.

4. The authors mention that the experiment with the transfer of mouse fibroblasts with isoform-specific coding plasmids revealed an enrichment of the fibroblast ECM but did not quantify that enrichment following the transformation. Also, it would be beneficial to support the presented data

by repeating the cell culture using EDA(+) and EDB(+) coated coverslips using an unaltered mouse fibroblast cell line.

We appreciate these suggestions. To demonstrate ECM enrichment of the fibroblast co-cultures, we now provide staining and quantification using our EDB- and EDA-specific antibodies (**Supplementary Fig. 2a-c**). The reason we resorted to myoblast coculture with mouse fibroblasts overexpressing the respective FN variants (kindly provided by Dr. Richard Hynes's lab through Addgene) was that we could not identify a reliable supplier for purified or recombinant full length mouse EDB(+)EDA(-) or EDB(-)EDA(+) FN. Thus, to address the reviewer's point, we resorted to decellularization of coverslips with fibroblast cultures after overexpression of the respective isoform-specific plasmids, and subsequently seeded myoblasts onto the ECMs (**Supplementary Fig. 2d**). This experiment revealed an effect similar to co-culture in which FN containing the EDB domain had positive effects on cell numbers, while the plasmids containing EDB(-)EDA(+) FN, pFN, or the empty vector, showed no difference (**Supplementary Fig. 2d-f**).

5. While investigating the mechanism of EDB(+)FN in myogenesis, the authors mention the correlation of p-Smad3 with Srsf1 but do not present correlation figures or a statistical approach used to quantify the results. Considering that in Fig. 5h-i, there is no significant difference between EDB(+) and vehicle (i), but Srsf1 expression is significant (h, vs. vehicle), it would also be interesting to see the correlation results of EDB(+)FN and Srsf1 expressions. What is the minimum expression of Srs1 that would support EDB(+)FN expression?

We thank the reviewer for pointing out this out. Most published studies use SIS3 in the μM range (IC50 of 3 μM). Thus, the amount of inhibitor we used (1-5 nM) is very low. However, we also wondered why only the 5 nM concentration of Smad3 inhibitor (SIS3) led to a statistically significant downregulation of both Srsf1 and EDB(+) FN compared to the vehicle control (**Fig. 5h,i**). At the same time, while 1 nM Smad3 inhibitor reduced Srsf1 expression by 20%, it did not significantly affect EDB(+) FN levels (although $p=0.08$ could be considered a trend). It is plausible that Smad3 inhibition at 1 nM SIS3 does not reduce Srsf1 levels sufficiently to affect EDB(+) FN splicing, while 5 nM blocks the pathway more potently. In addition, the expression analysis of Srsf1 and EDB(+) FN were performed at the same timepoint, 48h after application of SIS3. This suggests that the effects of reduced Srsf1 levels on EDB(+) FN splicing at 1 nM SIS3 could be more substantial after a longer incubation period. In light of these considerations, we decided to test a higher concentration of SIS3 (30 nM) and incubate the cells for a prolonged period of 72h. This experiment led to a 69% reduction in EDB(+) FN mRNA levels exceeding the results obtained for 5 nM SIS3 after 48h of incubation (37%) (**Supplementary Fig. 5a,b**). We also observed that 20 nM SIS3 in human myoblasts reduced Srsf1 expression by 72% and EDB(+) FN splicing by 90% after a 48h incubation (**Supplementary Fig. 5c,d**). Altogether, these results show that Srsf1 and EDB(+) FN levels correlate with the potency and duration of Smad3 inhibition. Interestingly, reduced levels of Srsf1 in response to Smad3 inhibition had to reach a certain threshold to affect EDB(+) FN production, suggesting that the involved splicing mechanism is highly efficient in activated MuSCs. We now discuss these new insights in the discussion section of our manuscript on **page 14, bottom of paragraph 3**.

6. It has been previously published that silencing Srsf1 reduces the inclusion of EDA isoform in fibronectin. Although the authors present the data that singles out the role of the EDB isoform, it would be interesting to see the expression of both isoforms upon SIS3 treatment.

Based on the publication by Lopez-Mejia et al. (Mol Biol Cell. 2013) referred to by the reviewer, we assessed EDA splicing after Srsf1 knockdown in myoblasts (**Supplementary Fig. 4d,e**). Similar to what was described in this paper, we observed a significant, but modest downregulation of EDA

splicing (7% reduction) after Srsf1 knockdown. Given the high affinity of the conserved Srsf1 binding motifs in the EDB sequence in the FN pre-mRNA (**Fig. 4a,b**) and the much more pronounced effects of knockdown on EDB splicing (62-78% reduction) (**Fig. 4d**), it is likely that a factor different from Srsf1 is responsible for the majority of EDA splicing in myoblasts. It is also important to keep in mind that there may be cell type-specific differences in Srsf1 activity and specificity. We now cite the paper by Lopez-Mejia and mention on these interesting considerations in the discussion of our paper on **page 15, paragraph 1**.

7. It is crucial to ascertain the expression of both isoforms to confirm the continuous primary role of EDB(+)-FN in MuSC differentiation. This comprehensive understanding is essential for advancing the field of research on sarcopenia.

We would like to highlight again that we made sure to not dismiss a potential role of EDA throughout our manuscript (please also see comments above). We analyzed the staining pattern of EDA(+)-FN (**Fig. 1f,h**) in skeletal muscle tissue and in the MuSC niche (**Fig. 1j**), studied EDA splicing in muscle tissue and different cell types in skeletal muscle (**Fig. 1c,d**), assessed the effects of exposure to EDB(-)-EDA(+)-FN on myoblasts (**Fig. 2a,b and Supplementary Fig. 2d-f**), studied EDA splicing upon Srsf1 knockdown (**Supplementary Fig. 4d,e**), and now also show that almost all EDB(+)-FN is in fact EDA(+) (**Supplementary Fig. 1c-e**). As discussed above, due to its specificity for MuSCs and its unique role in promoting myoblast proliferation, we focussed our present manuscript on the intricate up- and downstream pathways promoting EDB splicing and the novel cell autonomous mechanisms linked to it. To further contextualize related aspects of FN biology, we now also discuss potential roles of EDA, as well as of constitutive exons containing integrin-binding motifs such as RGD, on **page 15, paragraph 3** of our manuscript. This discussion covers potential direct roles of EDA and constitutive exons in regulating MuSCs as well as the possibility of synergistic effects with EDB.

8. The manuscript would be significantly enhanced by the inclusion of EDB-knockout mice, particularly aged mice. This addition would provide a more comprehensive understanding of the role of cellular fibronectin-specific isoforms during myogenesis.

We fully agree with the reviewer that analysis of EDB (and EDA) knockout mice, in particular a floxed allele that could be deleted in MuSCs and FAPs, would be insightful and complement our *in-vivo* siRNA knockdown data with cell type specificity. While we definitively plan to advance our work in this direction, our present manuscript is largely focussed on describing the novel TGF β -Smad3-Srsf1-EDB(+)-FN signaling pathway and its role for the auto-regulation of MuSCs. Sourcing and crossing of strains to generate cell-type specific knockout mice (and potentially age them) would take at least 2-3 years. While this is not feasible in the frame of this revision, we believe that our present manuscript will make an excellent foundation for realizing those goals in the future.

9. The paper would be more comprehensive with the inclusion of a limitation section in the discussion. This would help the authors acknowledge potential shortcomings and provide a more balanced view of their research.

We appreciate this suggestion. We included a paragraph describing limitations of our study in the discussion section on **page 15, paragraph 3** of our manuscript.

Reviewer #3 (Remarks to the Author):

We welcome this initiative by Nature Communications and appreciate the reviewer's contribution and constructive feedback.

REVIEWER COMMENTS

We sincerely thank our reviewers for their constructive and insightful feedback. In response, we have carefully revised our manuscript and addressed Reviewer #2's remaining comments, as detailed in our point-by-point response below. Specifically, (i) we expanded the discussion to provide a more comprehensive consideration of the EDA domain in the context of EDB(+) FN, (ii) refined formulations in the results section and in the legends for Fig. 6, (iii) added discussion of data regarding EDB(-)EDA(+) FN that were previously not included in the results section, (iv) included a statement on sample size and sex differences in the limitations paragraph of the discussion, and (v) added a paragraph on future directions and clinical significance. We believe these revisions have further strengthened the manuscript and hope it is now deemed suitable for publication in Nature Communications.

Reviewer #1 (Remarks to the Author):

I would like to thank the authors for their comprehensive and thoughtful revision. The additional data, methodological clarifications, and expanded discussion—particularly regarding the role of EDA(+) fibronectin and the inclusion of study limitations—have greatly improved the manuscript. The study provides novel insights into the TGF β –Smad3–Srsf1–EDB(+) FN signaling axis and its relevance in MuSC regulation and sarcopenia, and I believe it will be of strong interest to the field.

Congratulations on this valuable contribution.

We sincerely thank the reviewer for their thoughtful and encouraging comments. We are grateful for the recognition of our work and deeply appreciate the constructive feedback provided throughout the review process, which has greatly helped us strengthen our manuscript.

Reviewer #2 (Remarks to the Author):

The reviewer appreciates the authors' responses, their implementation of the suggestions, and their attention to the comments in the revised version. However, several concerns remain that require additional attention.

We appreciate the reviewers' helpful and constructive feedback, which has substantially strengthened our manuscript. We address the remaining comments below, with all changes in the manuscript file indicated in blue font.

1. The authors state in their reply “now also show that almost all EDB(+) FN is in fact EDA(+) (Supplementary Fig. 1c-e)”; therefore, how can they be certain that the observed effect is actually due to EDB(+) and not because the “transcripts containing EDB also contain the EDA domain in myoblasts”?

We thank the reviewer for highlighting the need for additional clarification regarding our conclusions on the role of the EDA extra domain. As noted in our previous responses, our study primarily focuses on EDB for two main reasons: (i) its unprecedented specificity for the activated MuSC niche, comparable to widely established stem cell markers such as m-cadherin and Pax7 (**Fig. 1e–h, Supplementary Fig. 1j**), and (ii) the pronounced pro-proliferative effects of ECM enriched in EDB(+)EDA(-) FN on myogenic progenitors, exceeding those of EDB(-)EDA(+) FN by several orders of magnitude (**Fig. 2a,b, Supplementary Fig. 2a–f**).

Accordingly, the core of our manuscript is devoted to the molecular pathways regulating EDB(+) FN splicing, which we found to minimally affect EDA(+) FN. For instance, Srsf1 knockdown reduced EDA inclusion by only ~7%, compared to 62–78% for EDB (**Fig. 4d, Supplementary Fig. 4d,e**). Importantly, the observation that nearly all EDB(+) transcripts also include EDA (**Supplementary Fig. 1c–e**), does not conflict with any of our conclusions. Rather, it suggests that EDA, given its

broad expression in skeletal muscle, likely serves more general biological roles but with less specificity to MuSCs.

Overall, our findings support the notion that EDB(+) FN, regardless of EDA status, is critical for MuSC function. The reasoning applied to EDA here similarly extends to other FN domains, such as the 9th and 10th FNIII domains (PHSRN motif and RGD site, respectively), which are also present in EDB(+) FN. These domains may have important, potentially synergistic effects with EDB, representing promising avenues for future investigation. Importantly, throughout the manuscript, we carefully avoided overstating conclusions or dismissing potential roles of EDA or other FN domains within EDB(+) FN. For instance, the graphical pathway summary in Fig. 8 explicitly depicts the presence of the EDA domain in EDB(+) FN transcripts and protein in MuSCs. Similarly, in the discussion's limitations section (**page 16, middle paragraph**), we acknowledge the potential for domain-domain interactions involving EDB:

“Although we discovered a novel autoregulatory mechanism controlling MuSC function through EDB(+) FN secretion and characterized the involved up- and downstream pathways, some limitations and outstanding questions of our study remain. The FN molecule contains several different domains with motifs that can engage different types of integrin heterodimers whose activation could have additive or synergistic effects during myogenesis. We observed that almost all EDB(+) FN also contains the EDA module, which raises the possibility that the two domains have essential complementary functions. Loss of FN containing the EDB domain might also affect integrin activation in myogenic cells indirectly due to lower availability of binding motifs in constitutive exons or the EDA domain. [...]”

Definitively dissecting the individual contributions of EDB, EDA, and other FN domains will ultimately require conditional MuSC-specific knockout mouse models using floxed alleles. While such an approach represents a substantial undertaking in terms of funding, breeding, and detailed phenotypic characterization, we believe our study establishes a strong foundation for these future investigations.

2. Figure 5 legend indicates, “myoblasts isolated from n=3 young mice per condition”, while the Figure 6a description states, “Principal component analysis (PCA) revealed that biological replicates from all three conditions showed distinct clustering (Fig. 6a)” and the legend describes “Data points and bars represent means from myoblasts isolated from n=3 (a-j) young mice per condition”. Figure 6a has only 3 data points. Therefore, both statements contradict each other. Altogether, the answer to the second comment remains unclear. Are there replicates or single animals? Is a single data point being presented as an average of 3 replicates from a single animal? Please clarify the following in the Statistical analysis, the main text, and the figure legends. Furthermore, if you have used 3 individual animals and had biological replicates, why cannot the sample size be increased by presenting all replicates of an individual animal or human used in the analysis?

We thank the reviewer for carefully noting this inconsistency. To clarify, unlike the bars in the graphs in **Fig. 6g,i,j**, which indeed represent mean values, each datapoint in the PCA plot in **Fig. 6a** represents a single measurement obtained from an individual biological replicate, with each replicate derived from a different mouse. We have corrected the figure legend (**page 41, last paragraph**) to read:

“Data points represent n=3 biological replicates per condition, each from an independent myoblast line isolated from separate young mice (a), and bars show means from n=3 biological replicates per condition, each from an independent myoblast line isolated from separate young mice (g,i,j).”

Throughout our study we deliberately present only biological replicates. This approach avoids artificially inflating sample size and ensures that our analyses reflect independent biological variability. As stated in the statistics section of the methods (**page 28, paragraph 1**):

“All datapoints in a given experiment and n numbers provided in the figure legends represent biological replicates each from cells or tissues of separate mice or humans.”

3. Please provide the representative images for Supplementary Fig. 1j,k.

To avoid repeating the same images, the representative panels corresponding to **Supplementary Fig. 1j,k** are shown in main **Fig. 1e–h**.

4. The authors haven't presented any correlation analysis. Therefore, the authors should refrain from using the term “correlation” and replace it with “associated”.

Thank you for bringing this to our attention. We replaced “correlated” with “associated” (**page 9, last paragraph**).

5. In their reply, the authors state that “the plasmids containing EDB(-)EDA(+) FN, pFN, or the empty vector, showed no difference”, but their figure demonstrates that EDB(-)EDA(+)FN is statistically different from the empty vector. Although the following statement is not written in the manuscript, this difference might play an important role in myogenesis.

We sincerely thank the reviewer for this attentive observation. The reviewer is absolutely correct - while the increase in cell numbers for EDB(-)EDA(+) FN compared with empty vector (EV) in **Supplementary Fig. 2f** is several magnitudes smaller (150%) relative to EDB(+)-EDA(-) FN compared with EV (525%), it is nonetheless statistically significant. We have updated the phrasing in the results section accordingly (**page 6, paragraph 2**):

*“We next co-cultured fibroblasts transfected with the different FN variants with freshly isolated MuSC-derived myoblasts from young mice expressing nuclear zsGreen under the Pax7 promoter²³. This experiment revealed that ECM containing EDB(+)-EDA(-) FN led to a 99%, 52%, and 60% increase in total Pax7-zsGreen positive myoblasts when compared to empty vector (EV), pFN or EDB(-)EDA(+) FN respectively (**Fig. 2a,b**). To exclude a confounding effect of cell-cell contacts or soluble factors, we also decellularized the fibroblast cultures after overexpression of the respective plasmids and seeded the ECMs with zsGreen positive myoblasts (**Supplementary Fig. 2d,e**). In line with the results obtained from fibroblast co-culture, this experiment revealed a 525%, 275%, and 150% increase in the number of myoblasts in the EDB(+)-EDA(-) FN condition when compared to EV, pFN or EDB(-)EDA(+) FN respectively (**Supplementary Fig. 2f**). We also observed a 150% increase in myoblast numbers in the EDB(-)EDA(+) FN condition relative to EV.”*

6. The Limitation section should include the small sample size and the absence of a sex difference investigation (using both male and female mouse samples).

We thank the reviewer for their thoughtful feedback. As noted in our Methods section under “Mice” (**page 20, paragraph 1**), we included male and female mice in approximately equal proportions for *in vivo* experiments (Fig. 3, n=4-5 mice per condition; Fig. 7, n=4-7 mice per condition). We did not observe sex differences, and thus samples from male and female mice were pooled. This is now mentioned under “Statistical Analysis” (**page 28, paragraph 1**):

“All datapoints in a given experiment and n numbers provided in the figure legends represent biological replicates each from cells or tissues of separate mice or humans. Since no sex differences were observed, data from male and female mice were pooled.”

We also wish to emphasize that all findings in our study were rigorously validated by using:

- i) complementary reagents (e.g., multiple siRNAs against the same target, or both mouse and human myoblasts),
- ii) multiple control conditions (e.g., empty vector, pFN, and EDB(-) FN),

- iii) complementary experimental approaches (e.g., myoblasts versus single fibers),
- iv) complementary readouts (e.g., cell counts and Ki67 staining, or activated integrin detection and itg β 31^{-/-} cells).

Effect sizes in *in vitro* experiments focused on EDB(+) FN splicing and its associated pathways were often substantial (often reaching two- to five-fold differences), ensuring adequate statistical power. For *in vivo* studies, and in alignment with 3R guidelines, our sample sizes (n = 4-7) are consistent with, and in several cases exceed, those commonly reported in academic publishing. Taken together, these considerations support the robustness of our findings, and we respectfully maintain that sample size alone does not constitute a major limitation of our study.

To accommodate the reviewer's comment, we have added the following statement to the limitations section of our manuscript (**discussion, page 16, end of paragraph 2**):

“/1..] Complementing these biological considerations, our in vitro experiments focusing on EDB(+) FN splicing and related pathways yielded robust effects, ensuring the experiments were adequately powered, and results were validated using complementary reagents and readouts in both mouse and human cells. Since no sex differences were observed, data from male and female mice were pooled. Nonetheless, more subtle sex-dependent effects cannot be excluded and may become apparent in studies with greater sample depth.”

7. What is the clinical significance and future directions of this study?

Thank you for raising this important point. We see the strongest potential for clinical significance in addressing the age-related decline in regenerative capacity, which contributes to poor outcomes due to inefficient healing of skeletal muscle injuries following invasive surgery or trauma in elderly patients. Future directions include targeting the pathways that regulate cell-autonomous EDB(+) FN signaling in MuSCs. In response to the reviewers' comment, we have added the following section to the discussion (**page 14, end of the middle paragraph**) where we elaborate on the regulation of the TGF β 31-Smad3-Srsf1-EDB(+) FN splicing pathway and its downstream effectors in aging:

“/1..] Direct therapeutic application of ECM molecules such as EDB(+) FN is inherently challenging, as in most cases they are large, poorly soluble, and possess a multimeric nature, which complicates their production, purification, and delivery. By uncovering the conserved TGF β 1-Smad3-Srsf1 splicing pathway and downstream integrin-mediated adhesion signaling as central regulators of cell-autonomous EDB(+) FN in the MuSC niche, we identify novel, actionable targets for therapeutic intervention. These insights lay the groundwork for in vivo modulation of this signaling axis, with the potential to enhance skeletal muscle repair and counteract age-related muscle decline.”

Reviewer #3 (Remarks to the Author):

We thank the reviewer for their contribution and welcome Nature Communications' initiative to recognize and support the training of Early Career Researchers in peer review.